# Chromatin conformation analysis of primary patient tissue using a low input Hi-C method

Noelia Díaz [1], Kai Kruse [1], Tabea Erdmann[2], Annette M. Staiger[3,4,5], German Ott[3], Georg Lenz[2] & Juan M. Vaquerizas [1]

Chromatin conformation constitutes a fundamental level of eukaryotic genome regulation. However, our ability to examine its biological function and role in disease is limited by the large amounts of starting material required to perform current experimental approaches. Here, we present Low-C, a Hi-C method for low amounts of input material. By systematically comparing Hi-C libraries made with decreasing amounts of starting material we show that Low-C is highly reproducible and robust to experimental noise. To demonstrate the suitability of Low-C to analyse rare cell populations, we produce Low-C maps from primary B-cells of a diffuse large B-cell lymphoma patient. We detect a common reciprocal translocation t(3;14)(q27;q32) affecting the *BCL6* and IGH loci and abundant local structural variation between the patient and healthy B-cells. The ability to study chromatin conformation in primary tissue will be fundamental to fully understand the molecular pathogenesis of diseases and to eventually guide personalised therapeutic strategies.

[1] Max Planck Institute for Molecular Biomedicine, Roentgenstrasse 20, 48149 Muenster, Germany. [2] Department of Medicine A, Hematology, Oncology and Pneumology, University Hospital Muenster, Albert-Schweitzer-Campus 1, Gebäude A1, 48149 Muenster, Germany. [3] Department of Clinical Pathology, Robert-Bosch-Hospital, Auerbachstrasse 110, 70376 Stuttgart, Germany. [4] Dr. Margarete Fischer Bosch Institute of Clinical Pharmacology, Auerbachstrasse 112, 70376 Stuttgart, Germany. [5] Eberhard Karls Universität Tübingen, Geschwister-Scholl-Platz, 72074 Tübingen, Germany. These authors contributed equally: Noelia Díaz, Kai Kruse. Correspondence and requests for materials should be addressed to J.M.V. (email: jmv@mpi-muenster.mpg.de)

The three-dimensional (3D) organisation of chromatin in the nucleus plays a fundamental role in regulating gene expression, and its misregulation has a major impact in developmental disorders[1,2] and diseases such as cancer[3]. The development of chromosome conformation capture (3C)[4] assays and, in particular, their recent high-throughput variants (e.g. Hi-C), have enabled the examination of 3D chromatin organisation at very high spatial resolution[5,6]. However, the most widely used current experimental approaches rely on the availability of a substantial amount of starting material—on the order of millions of cells—below which experimental noise and low sequencing library complexity become limiting factors[7]. Thus far, this restricts high-resolution analyses of population Hi-C to biological questions for which large numbers of cells are available and limits the implementation of chromatin conformation analyses for rare cell populations such as those commonly obtained in clinical settings. While single-cell approaches exist[8–11], they typically operate on much lower resolutions than population-based approaches and require an extensive set of specialist skills and equipment that might be out of reach for the average genomics laboratory.

Recently, two methods have been developed to measure chromatin conformation using low amounts of starting material[12,13]. However, the lack of a systematic comparison of the data obtained with these approaches and conventional in situ Hi-C limits our understanding of the technical constraints imposed by the amounts of starting material available. In addition, it remains to be demonstrated whether these methods could be directly applied to samples with clinical interest, such as for example, tumour samples.

Here, we present Low-C, an improved in situ Hi-C method that allows the generation of high-quality genome-wide chromatin conformation maps using very low amounts of starting material. We validate this method by comparing chromatin conformation maps for a controlled cell titration, demonstrating that the obtained maps are robust down to 1,000 cells of starting material and are able to detect all conformational features—compartments, topologically associating domains (TADs) and loops—similarly as maps produced with a higher number of cells. Finally, we demonstrate the applicability of Low-C to clinical samples by generating chromatin conformation maps of primary B-cells from a diffuse large B-cell lymphoma (DLBCL) patient. Computational analysis of the data allows us to detect patient-specific translocations and substantial amounts of variation in topological features.

## Results

### Low-C: A Hi-C method for low amounts of input material.
We first sought to develop a Hi-C method for low amounts of input material. To do so, we modified the original in situ Hi-C protocol[5], which recommends 5–10 million (M) starting cells, to allow for much smaller quantities of input material. The modifications are subtle, involving primarily changes in reagent volume and concentrations, as well as timing of the individual experimental steps (Fig. 1a, Methods, Supplementary Data 1). The combined changes, however, are highly effective, allowing us to produce high-quality Hi-C libraries from starting cell numbers as low as one thousand (1 k) cells.

To assess the feasibility and limitations of Low-C, we prepared libraries for progressively lower numbers of mouse embryonic stem cells (mESC) using two different restriction enzymes (Supplementary Table 1). Each library was deep-sequenced to an average depth of $100–150 \times 10^6$ reads and processed using a computational Hi-C pipeline with particular emphasis on the detection and filtering of experimental biases (Methods). The

ratios of the number of cis- and trans-contacts[14] indicate a high library quality for all samples (Methods) (Supplementary Table 2). Visual inspection of normalised Hi-C maps for 1 M to 1 k cells revealed a high degree of similarity between Low-C samples, with TADs clearly identifiable at a resolution of 50 kb (Fig. 1b, Supplementary Fig. 1a). To determine the degree of similarity between samples, we computed correlations of all contact intensities against the 1 M sample, which showed very high levels of reproducibility (Fig. 1b, Pearson correlation coefficient $R \geq 0.95$ in all cases). To evaluate the overall level of reproducibility with other protocols, we performed a comparison of a pooled Low-C dataset, merging samples up to 50 k cells, to a previously published mESC Hi-C dataset[15], to account for differences in sequencing depth. This comparison revealed a strong contact intensity correlation ($R = 0.97$), that was further confirmed by a principal component analysis that displayed strong clustering of Low-C samples and high similarity of Low-C to other mESC datasets (Supplementary Fig. 1b). In addition, we performed aggregate TAD and aggregate loop analysis[11] on the 1 k and 1 M samples (Fig. 1c, e), which revealed highly consistent TAD (Fig. 1d) and loop strengths (Fig. 1f) across datasets. Overall these results suggest that Low-C is a robust method to generate chromatin conformation maps using small amounts of input material.

### Low-C data have similar properties as conventional Hi-C.
We next wanted to ensure that the number of input cells does not limit the range of observations one can obtain from a Hi-C matrix. In a Hi-C experiment, each DNA fragment can only be observed in a single ligation product, limiting the number of possible contacts of the corresponding genomic region to twice the number of input cells (in a diploid cell line). This raises the concern for Low-C that low-probability contacts —such as those in far-cis—would be lost for very small numbers of cells. To test this, we calculated the correlation of contact intensities at increasing distances for the 100 k, 10 k, and 1 k against the 1 M sample. Reassuringly, while the expected decrease in correlation with distance was apparent, the decrease in contact correlation is independent of the input cell number (Fig. 2a), indicating that the loss of low-probability contacts was not a limiting factor for input cell numbers as low as one thousand. Furthermore, the remaining differences in correlation disappeared when comparing subsampled matrices to the same number of valid pairs (Supplementary Fig. 2a), suggesting that sequencing depth, and not the initial number of cells, is the main determinant of the correlation coefficients. We also confirmed that diversity of Hi-C contacts, measured as the absolute number of unique fragment pairs in a Low-C experiment, is not affected by the amount of input cells, but it is primarily a function of sequencing depth (Supplementary Fig. 2b-c).

To explore the limits of Low-C, we performed an extensive characterisation of the properties of these libraries. Previous work had identified systematic biases in Hi-C data that can serve as read-outs for the efficiency of Hi-C library generation[16–18] (Fig. 2b, Supplementary Fig. 3a-f). Most notably, PCR duplicates indicate low library complexity—a limitation that has been previously described when trying to scale down the Hi-C protocol[7]—while an excess of different types of ligation products, such as self-ligated fragments, can point to problems in the digestion and ligation steps (Methods). Unsurprisingly, given the higher need for amplification, we find that PCR duplicates increase with lower amounts of starting material, with roughly 20% of read pairs identified as duplicates in the 1 k sample (Fig. 2b). Ligation errors, however, remained more or less constant across samples, irrespective of the number of cells

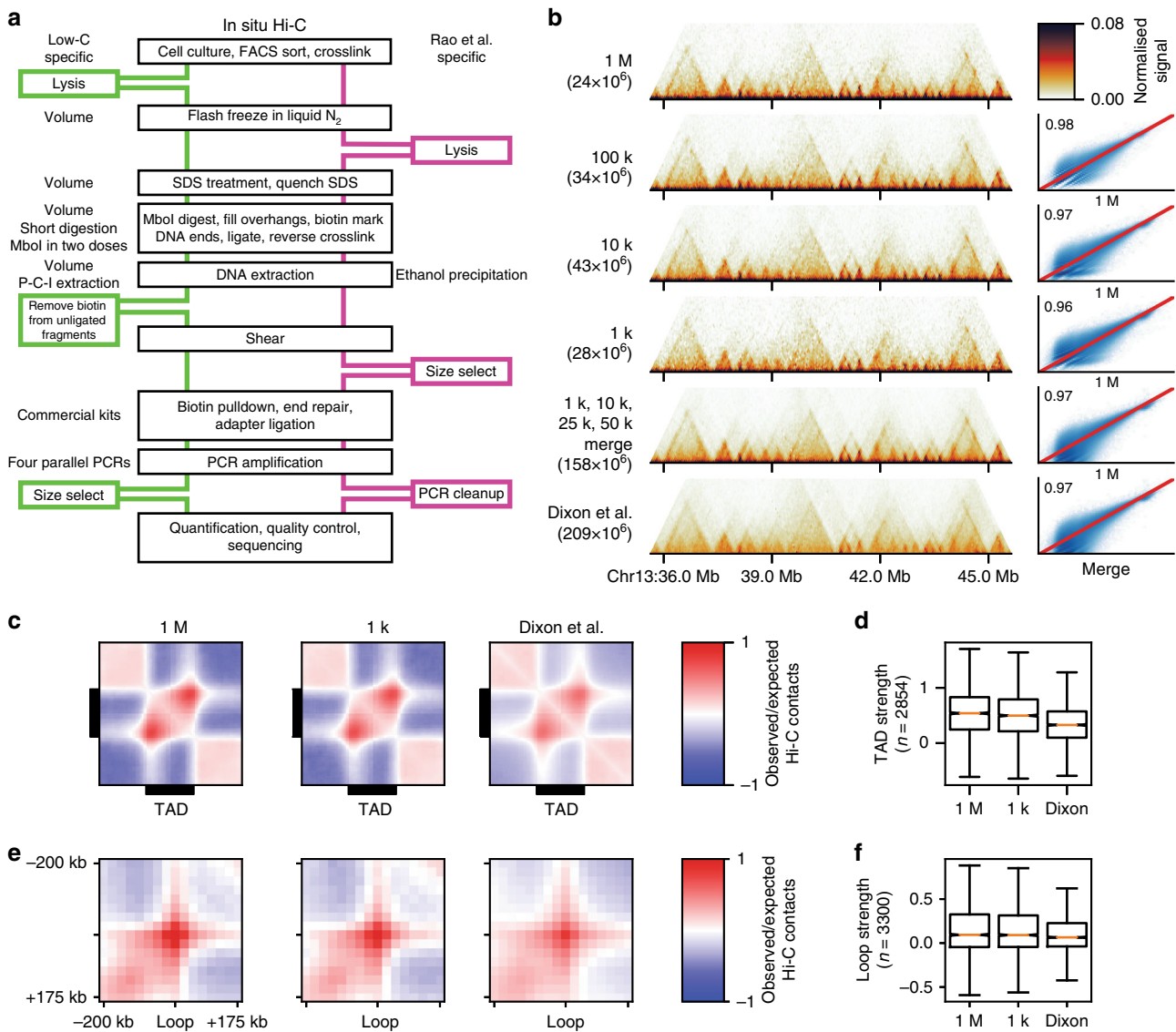

**Fig. 1** Low-C enables the examination of chromatin architecture for samples with low amounts of input material. **a** Schematic overview of the Low-C protocol and comparison with the previously published in situ Hi-C protocol from Rao et al.[5]. Black boxes denote common steps in both protocols. Green and magenta boxes denote additional steps in the Low-C and in situ Hi-C protocol, respectively. Italicised text marks protocol-specific differences regarding the step next to it. (P–C–I = Phenol–Chloroform–Isoamyl alcohol). **b** Low-C matrices for a 10 Mb region on chromosome 13. Input cell numbers for the Hi-C matrices shown span four orders of magnitude (1 M to 1k cells). Pixel intensity corresponds to normalised counts. The bottom two Hi-C matrices display data from a merge of Low-C samples (1k, 10k, 25k, and 50k), and a previously published ESC dataset from Dixon et al.[15] with similar sequencing depth as the merged sample. In brackets below the sample label, we list the number of valid read pairs in each Hi-C library. Next to the Hi-C matrices scatter plots and Pearson correlation coefficient of the contact intensities in the 50 kb resolution maps of each sample on the left against the 1 M sample are shown. The correlation and scatter plots next to the Dixon et al.[15] dataset correspond to a comparison with the merged sample. Red line indicates identity. **c** Aggregate TAD analysis of the 1 M, 1k, and Dixon et al.[15] Hi-C maps. Shown is the average observed/expected ratio of Hi-C signal for regions around all TADs as determined by Rao et al.[5]. **d** Comparison of TAD strength (Methods) for the 1 M, 1k, and Dixon et al.[15] samples. **e** Aggregate loop analysis showing the average observed/expected Hi-C signal at all loop regions as determined by Rao et al.[5]. **f** Comparison of loop strength (Methods) for the 1 M, 1k, and Dixon et al.[15] samples. **d**, **f** Boxes span the interquartile range (IQR), i.e. they extend from the first (Q1) to the third quartile (Q3) values of the data, with a line at the median. Whiskers span [Q1 − 1.5 × IQR, Q3 + 1.5 × IQR], outliers are omitted

(Supplementary Fig. 4). Other low-input Hi-C datasets[12,13] display similar biases (Fig. 2a, b), confirming that decreasing library complexity appears to be the strongest limitation on the lowest number of input cells that is feasible for low-input Hi-C approaches.

**Compartments, TADs and loops can be detected in Low-C data.** Next, we set out to ensure that not only the Hi-C maps

themselves, but also measures derived from them are reproducible and unaffected by differences in input cell number. To do so, we calculated several established and widely used Hi-C measures on the Low-C matrices at 50 kb resolution, namely: the profile of expected contacts at increasing distances between genomic regions[6] (Fig. 3a, Supplementary Fig. 5); the correlation matrix and its first eigenvector, used to derive AB compartments[6] (Fig. 3b, Supplementary Fig. 6); and the insulation score[19], commonly used to infer TADs and TAD boundaries[20] (Fig. 3c).

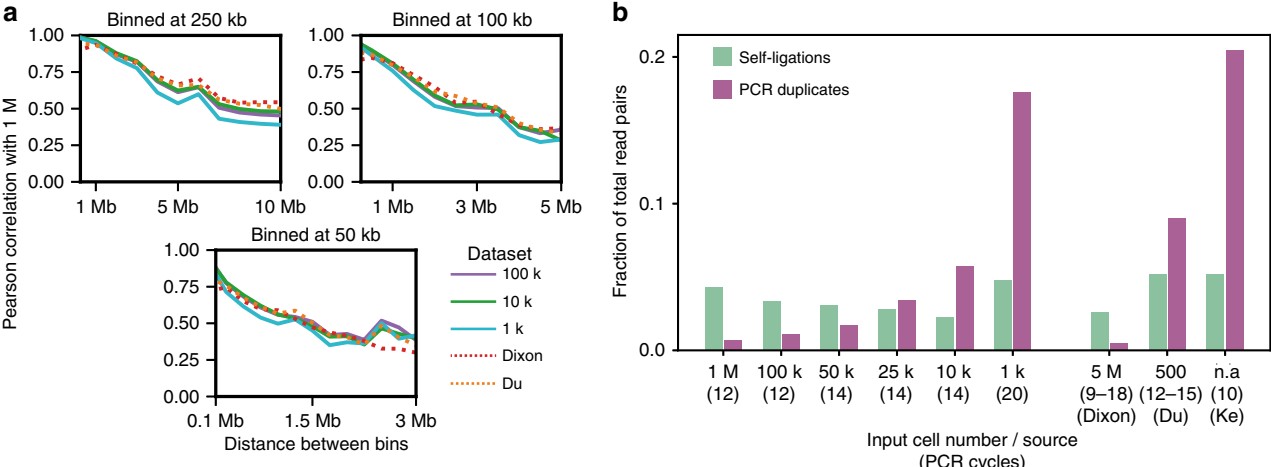

**Fig. 2** Analysis of experimental biases and quantitative properties of Low-C libraries. **a** Pearson correlation coefficient for contact intensities in bin pairs at increasing distances from the diagonal. Each plot represents different bin sizes, indicated above the plot. Colours correspond to input cell number or sample source, respectively. All reported correlations are with the 1 M sample. **b** Fraction of fragment pairs affected by and filtered out due to self-ligated fragments or PCR duplicates. Input cell numbers and PCR cycles (brackets) are indicated on the bottom of the plot. Note that all samples are in mESC, except for the Ke et al.[13] sample on the right, which is in zygote (PN5)

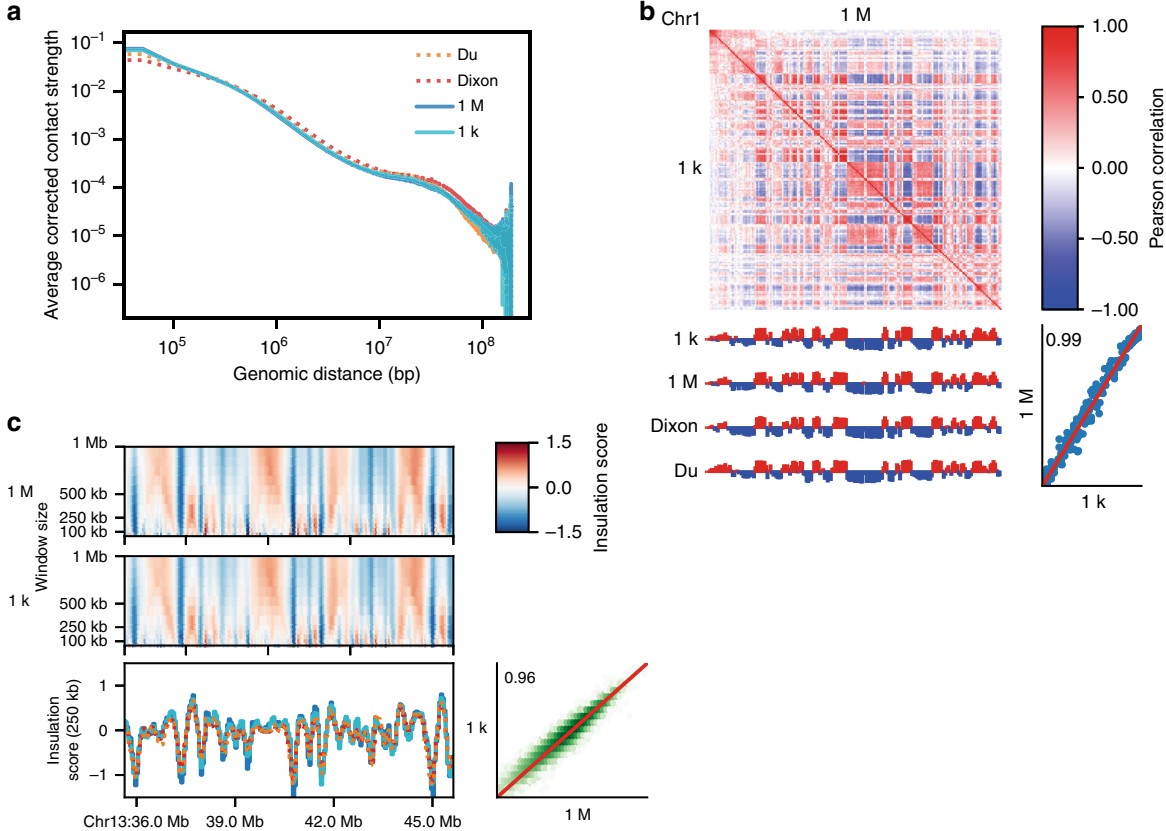

**Fig. 3** Compartments, TADs and loops can be detected and are highly reproducible in Low-C data. **a** Log–log 'distance decay' plot for chromosome 1 showing the decrease in contact probability between 50 kb bins with increasing distance for the 1 M and 1 k, as well as the Dixon et al.[15] and Du et al.[12] samples. **b** AB compartment comparison for chromosome 1 binned at 1 Mb. Contact correlation matrices for the 1 M and 1 k samples (top) and the corresponding first eigenvector (coloured according to the sign of the eigenvector entries) for the samples listed in **a** are shown on the left. Bottom right shows a scatter plot of first eigenvector values of the 1 M vs. the 1 k sample with Pearson correlation coefficient shown in the plot. Red line indicates identity. **c** Insulation score[20] comparison for the region on chromosome 13 shown in Fig. 1b. Heatmaps display insulation score values for a range of window sizes, line plots highlight the insulation index for a window size of 250 kb (see panel **a** for line colours). Next to it is a scatter plot with the Pearson correlation coefficient of the complete insulation index vectors for the 1 k and 1 M samples. Red line indicates identity

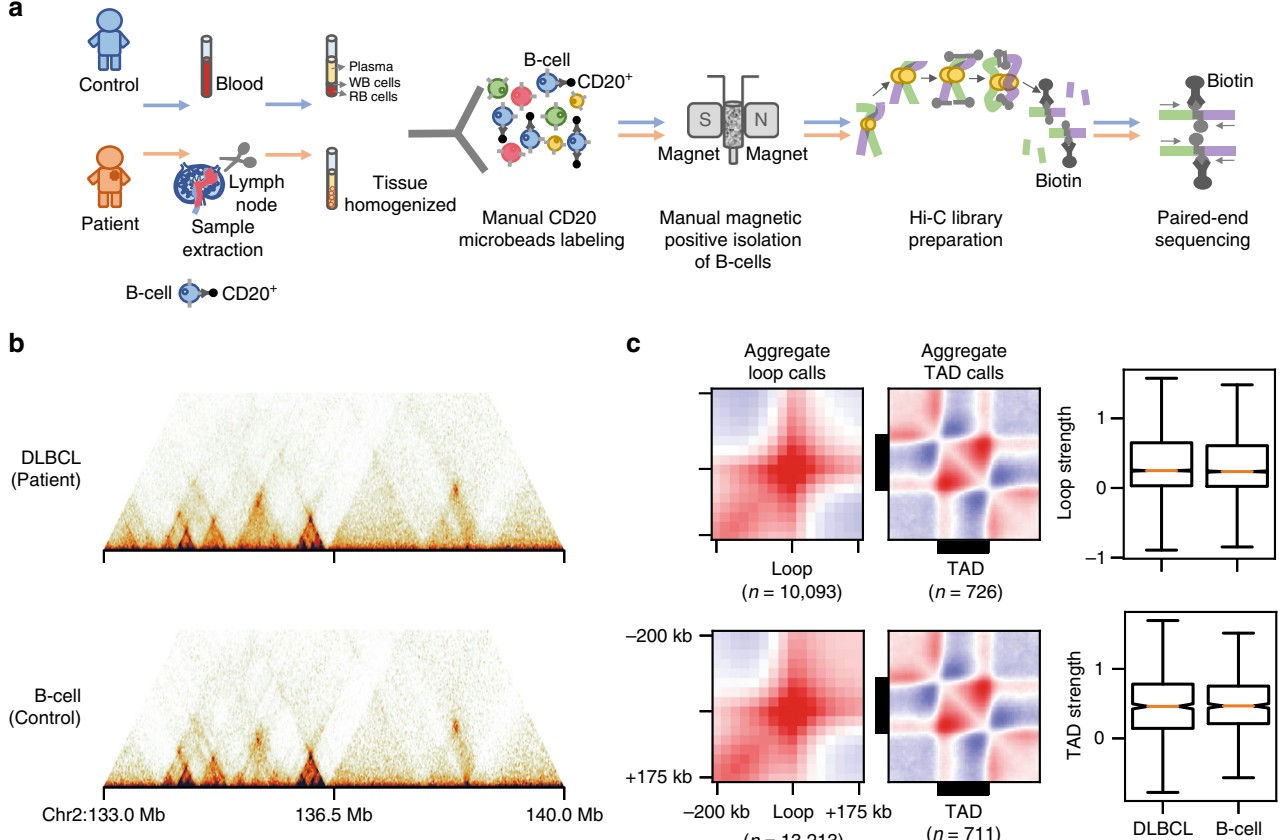

**Fig. 4** Generation of Low-C maps for DLBCL primary tissue and healthy B-cells. **a** Schematic overview of DLBCL (patient) and healthy B-cell (control) sample extraction using MACS sorting of CD20$^+$ microbead-labelled cells extracted from primary tissue and Low-C library generation. **b** Sample region on chromosome 2 highlighting TADs, loops and loop domains in the Low-C maps for DLBCL and healthy B-cells. **c** Aggregate TAD and loop analysis (see Fig. 1) using de novo loop and TAD boundary calls (Methods). On the right is a comparison of loop and TAD strengths between patient and control. Boxes span the interquartile range (IQR), i.e. they extend from the first (Q1) to the third quartile (Q3) values of the data, with a line at the median. Whiskers span [Q1 − 1.5 × IQR, Q3 + 1.5 × IQR], outliers are omitted

All three examples of Hi-C measures are consistent with results from conventional Hi-C and showed high reproducibility between the 1 M and 1 k samples with no apparent dependence on the number of input cells, demonstrating that Low-C libraries are highly consistent and reproducible for input cell numbers as low as 1,000 cells.

**Generation of Low-C maps for DLBCL primary tissue.** Given our ability to obtain high-quality chromatin conformation maps using low amounts of input material, we sought to determine whether the technique could be applied in a real-world scenario where the amount of starting material is likely to be the limiting factor in obtaining chromatin contacts maps. To test this, we performed Low-C on a diffuse large B-cell lymphoma (DLBCL) sample and in normal B-cells extracted from a healthy donor as a control (see Methods). Generating chromatin contact maps with low amounts of input material is beneficial not only because it allows to test the 3D chromatin conformation directly in the diseased cells, but also since it maximises the availability of tissue for other procedures and minimises patients' burden from having to undergo repeated biopsies to obtain extra material.

Patient and donor CD20$^+$ lymphocytes were isolated from lymph nodes and blood, respectively, using a magnetic microbead-labelled CD20$^+$ antibody and magnetic-activated cell sorting (MACS)[21] (Fig. 4a, Methods). We confirmed that the cell fixation procedure did not affect the efficiency of MACS sorting and that we were able to correctly distinguish CD20$^+$

from CD20$^-$ cells in a mixture of HBL1 and Jurkat cells (Supplementary Fig. 7) and in the peripheral blood mononuclear cells (PBMCs) from the control sample (Supplementary Fig. 8), respectively before and after formaldehyde fixation. Using the MACS approach, we were able to isolate the majority of B-cells from the control sample and the cell line mixture, although a non-critical fraction of B-cells was lost during the process (Supplementary Fig. 9). The same was true for the patient sample where fixation did not affect the surface molecules needed for MACS sorting (Supplementary Fig. 10) and where the eluted CD20$^+$ cell population was made up of 95.5% B-cells (Supplementary Fig. 10f). We then performed Low-C on ~50 k cells from each of the patient and control samples and deep-sequenced the resulting libraries to approximately 500 million (patient) and 300 million (control) reads (Supplementary Table 1). The resulting chromatin maps show a high degree of similarity between the patient and control B-cells (Fig. 4b). TADs (ordinary and loop domains) and loops are clearly distinguishable in the maps, and de novo loop calling using HICCUPS[5] and subsequent aggregate loop analysis (Fig. 4c) confirms that these can be identified automatically with high confidence. Overall these results confirm that Low-C can be successfully used in a clinical setup to obtain high-quality chromatin conformation maps directly from primary patient tissue.

**Identification of structural variation in patient Low-C data.** Structural variation and, in particular, genome rearrangements

are a characteristic feature in many cancers[22]. Since chromatin contact maps have an intrinsic bias for detecting interactions that happen in the proximal linear sequence[6], Hi-C-like data can be used to detect structural variation[1–3,23–28]. In order to detect potential translocations in the DLBCL sample in a fully automated and unbiased manner, we performed virtual 4C (V4C) for the patient and control data. Specifically, we considered each 25 kb bin in the genome in turn as a viewpoint to detect cases that display significant amounts of signal anywhere in the genome of the DLBCL cells that do not appear in control B-cells (Methods). Hi-C maps at locations of putative structural variations were then browsed manually to remove false positives.

Most prominently, this analysis identified two regions of interest on chromosome 3q27 separated by ~8 Mb, with significant interactions with chromosome 14q32 (Fig. 5a–c). As expected, a normal V4C profile was observed around the viewpoint in chromosome 3 for both regions in both control and DLBCL cells. In contrast, the V4C profile found for the interacting regions on chromosome 14q32 was only apparent in the patient data, suggesting that the interactions are patient-specific. A closer examination of the genes located in the patient-interacting regions revealed that the first viewpoint (Fig. 5, magenta shaded region) lies directly at the BCL6 gene, a transcription factor known to be affected in DLBCL, while the interacting region on chromosome 14q32 lies at the immunoglobulin heavy-chain (IGH) locus (Fig. 5d, e), suggesting a t(3;14)(q27;q32) reciprocal translocation. Translocations involving BCL6 are among the most commonly observed rearrangements in DLBCL[29–31], with one study reporting a ~30% (14/46) penetrance in DLBCL patients[32]. The second viewpoint with significant interactions towards the telomeric end of chromosome 3 (Fig. 5, green shaded region) interacts with a more centromeric location on chromosome 14. The pattern of interaction signal decay over linear distance in the trans-chromosome interactions map suggests a breakpoint around 195.2 Mb (Fig. 5d, f, black triangles) and allows us to manually reconstruct the most likely rearrangement of these regions in DLBCL from the Hi-C data: the telomeric ends of both chromosomes are involved in a reciprocal translocation, with breakpoints around Chr3:187.7 Mb and Chr14:105.9 Mb (Fig. 5h). To validate our data, we performed a fluorescence in situ hybridisation (FISH) analysis that confirmed a rearrangement of the BCL6 gene (Fig. 5i), providing orthogonal validation of the Hi-C findings. In addition, the lack of Hi-C signal between the breakpoints in chromosome 3 and 14 suggests that the regions Chr3:187.7-195.2 Mb and Chr14:105.6-105.9 Mb have been lost on one pair of chromosomes, generating regions of loss of heterozygosity in the remaining chromosome. Interestingly, we find another smaller rearrangement involving ANXA3 on chromosome 4 and EDAR2 on chromosome X (Supplementary Fig. 11). Misregulation of ANXA3 is known to promote tumour growth, metastasis and drug resistance in both breast cancer[33] and hepatocellular carcinoma[34]. In summary, our results demonstrate that Low-C can be used directly on primary tissue to detect patient-specific chromosomal rearrangements in an unbiased manner.

**Extensive rewiring of chromatin organisation in DLBCL cells.**
Visual comparison of the patient and control chromatin contact maps revealed numerous local structural differences. For example, the region undergoing loss of heterozygosity reported above (Chr3:187.7–195.2 Mb; Fig. 5d, arrow) displays a clear gain of TAD structure encompassing the genes TP63, a member of the p53 family of transcription factors that has been previously associated with cancer[35], and the tumour protein p63 regulated gene-1 (TPRG1) which lies in the same de novo established TAD,

suggesting their potential co-regulation. To evaluate the overall extent of changes in chromatin conformation at the TAD structure level between the two samples, we used the insulation score[19] to determine TAD boundaries in both samples and looked for regions with broad changes in the Hi-C signal (Methods). Using a conservative threshold, we detected 648 regions in the genome with notable changes in local Hi-C contacts (Supplementary Data 2). Out of these, 37 appear to be de novo TADs, which in many cases overlap with known disease-related genes such as PTPRG[36] (Fig. 6c), APBB2[37] (Fig. 6d), and TEAD1[38,39] (Fig. 6e). Overall, we observe the majority of changes in TAD structure to be patient-specific gains, whereas the loss of TADs present in normal B-cells in the patient is a relatively rare event (Fig. 6b). Altogether, our results demonstrate that Low-C can be used to study chromatin contact differences between patient samples at the TAD level and that there are significant differences in TAD structure between DLBCL and normal B-cells.

## Discussion

The development of high-throughput genome-wide techniques to measure chromatin conformation has been instrumental to further our understanding of the biological importance of the three-dimensional organisation of chromatin in the nucleus. In addition to providing a local environment where enhancer-promoter interactions can orchestrate the correct deployment of gene expression programmes during development, the 3D chromatin conformation is fundamental to establish proper spatial boundaries, that provide enhancer insulation and limit their function to those genes that need to be regulated. Chromatin conformation at the level of TADs seems to be fairly static for fully differentiated cells[15,40,41], although dynamic changes in TAD structure can be observed during development in organisms ranging from Drosophila to mammals[12,13,42,43], highlighting their dynamic behaviour.

A current limitation for our understanding of these dynamic changes and the potential differences in 3D chromatin conformation between tissues or in a disease context is the high amount of material that is usually necessary to perform these experiments. While single-cell Hi-C methods exist, these are usually only able to capture a small fraction of the chromatin contacts that occur across the genome. This results in sparse chromatin maps of low resolution that usually rely on TAD calls made using standard Hi-C maps, limiting their applicability in comparing samples or finding de novo TADs.

Here, we introduce Low-C, an improved Hi-C method that allows the generation of high-resolution chromatin contact maps using low amounts of input material. Beyond existing low input Hi-C approaches[12,13], we perform a thorough comparison of Low-C maps and their derived measurements in a controlled environment to systematically demonstrate that Low-C is not affected by biases originating from the amount of starting material. We also show that the method is robust and applicable to mammalian samples down to 1,000 cells without compromising the quality of the resulting datasets. Therefore, our results establish Low-C as an efficient method to study chromatin conformation for rare cell populations, where the collection of material currently necessary to perform population-based Hi-C protocols is infeasible. These include transient developmental stages[12,13,42], as well as systems of medical relevance, such as primary tissue from patient samples, where an examination of changes in chromatin conformation between healthy and disease cells might shed light on the aetiology of the disease.

To demonstrate the usability of this approach in a real-world scenario, we generated Low-C maps for a DLBCL patient sample. Since changes in chromatin contact profiles and genomic

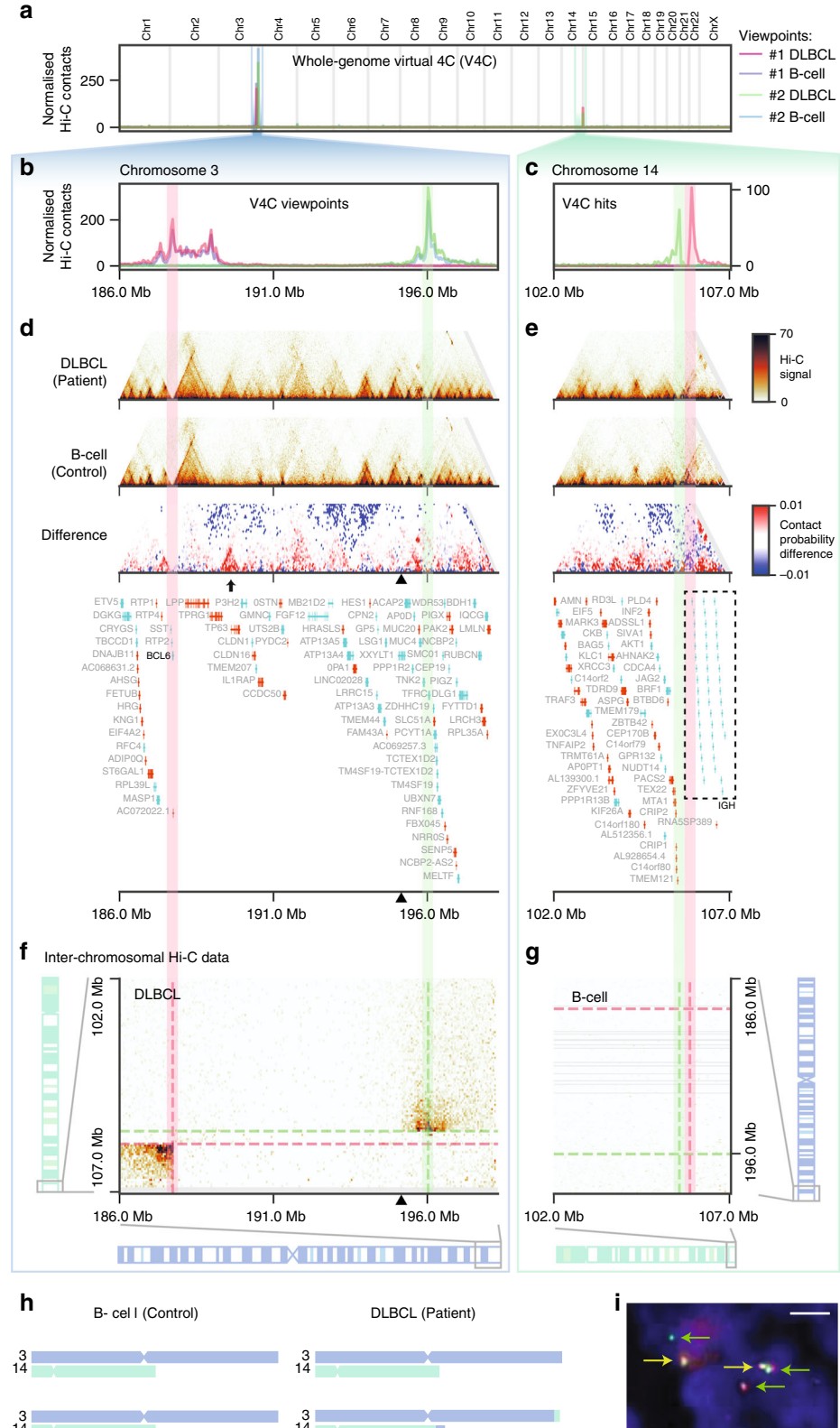

**Fig. 5** Unbiased detection and characterisation of t(3;14)(q27;q32) reciprocal translocation in the DLBCL sample. **a** Whole-genome virtual 4C plot for two viewpoints on chromosome 3 (magenta, green). **b, c** Zoom-in of the virtual 4C plots to the viewpoint (**b**) and target (**c**) regions. **d, e** Local Hi-C maps of the viewpoint (**d**) and target (**e**) regions. **f, g** Inter-chromosomal Hi-C maps of target vs. viewpoint region in the patient (**f**) and viewpoint vs target region in control (**g**), highlighting the telomeric ends of the corresponding chromosomes. Inter-chromosomal Hi-C maps saturate at a signal intensity of 20, rather than 70 as in **d** and **e**. **d, f** Black triangles at the x axis indicate most likely breakpoint region as determined visually from the inter-chromosomal Hi-C maps in **f**. **h** Schematic representation of the reciprocal translocation as interpreted from the Hi-C data. **i** Confirmation of *BCL6* translocation by FISH using a *BCL6* Dual Colour, Break Apart Rearrangement Probe hybridised to the patient's tumour sample. A translocation of the *BCL6* gene is indicated by separated red and green signals (green arrows). In addition, two non-rearranged red/green fusion signals are seen (yellow arrows). Scale bar: 3.1 μm

rearrangements can be detected very easily through Hi-C approaches[44], we developed an unbiased approach to systematically detect translocations using these data, uncovering a known reciprocal translocation in this patient biopsy. This, together with recent reports of similar approaches in other tumour types[23] highlights the clinical applicability of this technology. An added benefit of our approach when compared with previous work in primary tissue samples is the generation of high-quality genome-wide chromatin interaction maps, which allows us to examine the level of variability between cells in health and disease. In fact, we detect a large amount of variation at the TAD level, in particular in the DLBCL sample, which gains a significant amount of structure. Interestingly, in several cases the emergence of novel chromatin structural features coincides with the genomic location of genes previously associated with cancer, such as TP63 and ANXA3. Whereas the current maps do not allow us to determine cause or consequence for these changes, a broader examination of these changes in larger cohorts of patient samples, together with an integrative analysis of gene expression and chromatin states might provide insight into the causal relationships between these in a disease-specific and patient-specific manner.

Despite the increased applicability of our method, there are still a number of factors to take into consideration when planning such experiments. First, tissue heterogeneity or the presence of healthy cells in biopsies can become an issue with increasingly lower cell numbers. Specifically, the lower the input cell number, the greater the impact of contaminations or variabilities in sample composition will be on the averaged chromatin structures visible in the Hi-C maps. These might obfuscate or increase the uncertainty about specific structural observations. In our DLBCL analysis, we set out to minimise these effects by coupling our Low-C to efficient cell sorting techniques. Second, decreasing library complexity is still the current limiting factor for low input Hi-C studies[7], and a significant amount of PCR duplicates are to be expected when reducing the amount of starting material. Third, a further general limitation for bulk Hi-C methods, regardless the initial cell input, is that long-range three-dimensional contacts between gene promoters and enhancers are likely to be missed, since they usually happen within the context of TAD interactions. Therefore, to study these important interactions, which have been shown to affect gene regulation and are associated with the risk for various types of diseases[45,46], it might be useful to couple Low-C with capture or promoter-capture techniques[47–50], that will allow the retrieval of these specific interactions.

In summary, our data demonstrates that it is feasible to obtain high-quality genome-wide chromatin contact maps from low amounts of input material. We anticipate that the robustness and relatively simple implementation will make Low-C an attractive option that will facilitate bringing the analysis of chromatin architecture within reach of personalised clinical diagnostics.

## Methods

**Low-C protocol**. We followed the general protocol for in situ Hi-C as described in Rao et al.[5], which we adapted for use on low cell numbers. Mainly, differences were related to adjustments in the volume of the reactions, a shortening of the digestion step, a removal of biotin from the unligated fragments, and an alternative strategy for size-selection during library preparation (Fig. 1a, Supplementary Data 1), as described below.

**Cell culture**. mESC OG2 cells were cultured as described in Shi et al.[51]. Briefly, mouse embryonic stem cells derived from OG2 mice (B6; CBA-Tg(Pou5f1-EGFP) 2Mnn/J; stock number 004654) were maintained in high-glucose (4,500 mg/l) Dulbecco's modified Eagle's medium (DMEM; Sigma-Aldrich), supplemented with N2 supplement 1 mL/100 mL media (Gibco, 17502-048), 1 μM MEK inhibitor PD0325901 (Gibco), 3 μM GSK3 inhibitor CHIR99021 (Gibco), 15% serum replacement (Gibco), 2% fetal bovine serum (FBS; Biochrom), 2 mM L-Glutamine

(Sigma), 1% penicillin/streptomycin (Sigma-Aldrich, P4333), 1% non-essential amino acids (Sigma-Aldrich), 0.10 mM beta-mercaptoethanol (Gibco) with homemade leukemia inhibitory factor (LIF; 1,000 units LIF/mL) grown on a 0.1% gelatin coated plates and feeder-free. Cells were FACS-sorted, selected for positive eGFP expression and collected in PBS. Cells were then pelleted (300 × g, 4 °C for 10 min) and resuspended in 1 mL PBS.

HBL1 (B-cells; Cellosaurus accession number CVCL_4213) and Jurkat (T-cells; Cellosaurus accession number CVCL_0065) cells from frozen stocks were used to test the resilience of surface molecules after formaldehyde fixation, as well as for testing the capacity of the MACS protocol for pure B-cell population isolation.

**Cross-linking**. Cells were then cross-linked in a 1% final concentration (v/v) of 37% formaldehyde (VWR International GmbH) prior to 10 min room temperature incubation with gentle rotation (20 rpm). The reaction was quenched for 5 min at room temperature with gentle rotation (20 rpm) by adding 2.5 M Glycine solution (Applichem) to a final concentration of 0.2 M. Cells were pelleted twice (300 × g, 4 °C for 5 min) and resuspended on 1 mL of cold 1× PBS. After these washes, cells were finally resuspended in 50 mL, quantified on a Neubauer's chamber and aliquoted at different cell concentrations (5 × 10^6, 10^6, 10^5, 5 × 10^4, 2.5 × 10^4, 10^4, and 10^3 cells).

**Lysis**. Cells were pelleted (300 × g, 4 °C for 5 min) and gently resuspended in 500 μL (5 M and 1 M samples), 250 μL (100 k, 50 k, 25 k, and 10 k samples), or 125 μL (1 k sample) of ice-cold in situ Hi-C lysis buffer (10 mM Tris-Cl pH 8.0, 10 mM NaCl, 0.2% IGEPAL CA-630 (Sigma-Aldrich), cOmplete Ultra protease inhibitors (Roche)). Following 15 min incubation on ice, cells were spun down (1000 × g, 4 °C for 5 min) and the pellet was resuspended in 250 μL (5 M and 1 M samples), 125 μL (100 k, 50 k, 25 k, and 10 k samples) or 62.5 μL (1 k sample) of ice-cold in situ Hi-C lysis buffer. Lysed cells were then flash-frozen in liquid Nitrogen and stored at −80 °C.

**Restriction enzyme digestion**. Frozen aliquots for different cell concentrations were placed on ice to thaw, spun (300 × g for 5 min at 4 °C) and resuspended in 500 μL (5 M and 1 M samples), 250 μL (100 k sample), 125 μL (50 k, 25 k, and 10 k samples), or 62.5 μL (1 k sample) lysis buffer. Following yet another spin (13,000 × g for 5 min at 4 °C), cells were gently resuspended in 500 μL (5 M and 1 M samples), 250 μL (100 k sample), 125 μL (50 k, 25 k, and 10 k samples) or 62.5 μL (1 k sample) ice-cold 10× NEB2 buffer (New England Biolabs). Then, nuclei were spun once more for 5 min at 13,000 × g at 4 °C before being permeabilised by resuspending them in 50 μL (5 M, 1 M and 100 k samples), 25 μL (50 k, 25 k, and 10 k) or 12.5 μL (1 k sample) of 0.4% SDS and incubating for 10 min at 65 °C without agitation. The SDS (Applichem) was then quenched by adding 25 μL (5 M, 1 M and 100 k samples) or 12.5 μL (50 k, 25 k, 10 k, and 1 k samples) of 10% Triton X-100 (Applichem) and 145 μL or 72.5 μL of nuclease-free water, respectively, and incubating at 37 °C for 45 min with rotation (650 rpm). Chromatin digestion was performed as follows: (i) for HindIII digested samples, by adding 500 U of New England Biolabs HindIII-HF (5 M and 100 k samples) in 25 μL of 10× NEB2 buffer; and (ii) for MboI (New England Biolabs) digested samples, by adding 100U (1 M and 100 k samples), 50 U (50 k, 25 k, and 10 k samples), or 25 U (1 k sample) of MboI in 20 μL (5 M, 1 M and 100 k samples), 35 μL (50 k, 25 k, and 10 k samples) or 42.5 μL (1 k sample) of 10× NEB2.1 buffer (New England Biolabs), respectively. All digestions were performed at 37 °C with gentle rotation for a period of 90 min by adding the restriction enzyme in two instalments, the second one after 45 min. After digestion, only MboI samples were heat-inactivated for 20 min at 62 °C.

**Marking of DNA ends**. The overhangs generated by the restriction enzyme cuts were filled-in by adding a mix of biotin-14-CTP (0.4 mM stock; 18.75 μL for the 5 M to 100 k samples or 10 μL for the 50 k to 1 k samples; Life Technologies), 10 mM dATP (whichever was not supplied in biotinylated form), dGTP and dTTP (10 mM stocks; 0.75 μL of each dinucleotide for the 5 M to 100 k samples or 0.5 μL for the 50 k to 1 k samples), and 5 U/μL DNA polymerase I Klenow (5 M to 100 k samples or 4 μL for the 50 k to 1 k samples; New England Biolabs), followed by a 90 min incubation at 37 °C with gentle rotation (20 rpm).

**Proximity ligation**. The resulting DNA fragments were then ligated by adding a master mix containing nuclease-free water (657 μL for the 5 M to 100 k samples or 328.5 μL for the 50 k to 1 k samples), 10× T4 DNA ligase buffer (120 μL for the 5 M to 100 k samples or 60 μL for the 50 k to 1 k samples; Thermo Fisher Scientific), 10% Triton X-100 (100 μL for the 5 M to 100 k samples or 50 μL for the 50 k to 1 k samples), 20 mg/mL BSA (12 μL for the 5 M to 100 k samples or 6 μL for the 50 k to 1 k samples; New England Biolabs) and 5 Weiss U/μL T4 DNA ligase (5 μL for the 5 M to 100 k samples or 3.5 μL for the 50 k to 1 k samples in two instalments; Thermo Fisher Scientific). Samples were mixed by inversion and incubated at 20 °C with gentle rotation (20 rpm) for 4 h.

**Crosslink reversal**. Nuclei were then spun (2500 × g for 5 min at room temperature) and resuspended in 500 μL (5 M to 100 k samples) or 250 μL (50 k to 1 k samples) extraction buffer to revert the crosslinking. To digest the proteins 20 μL

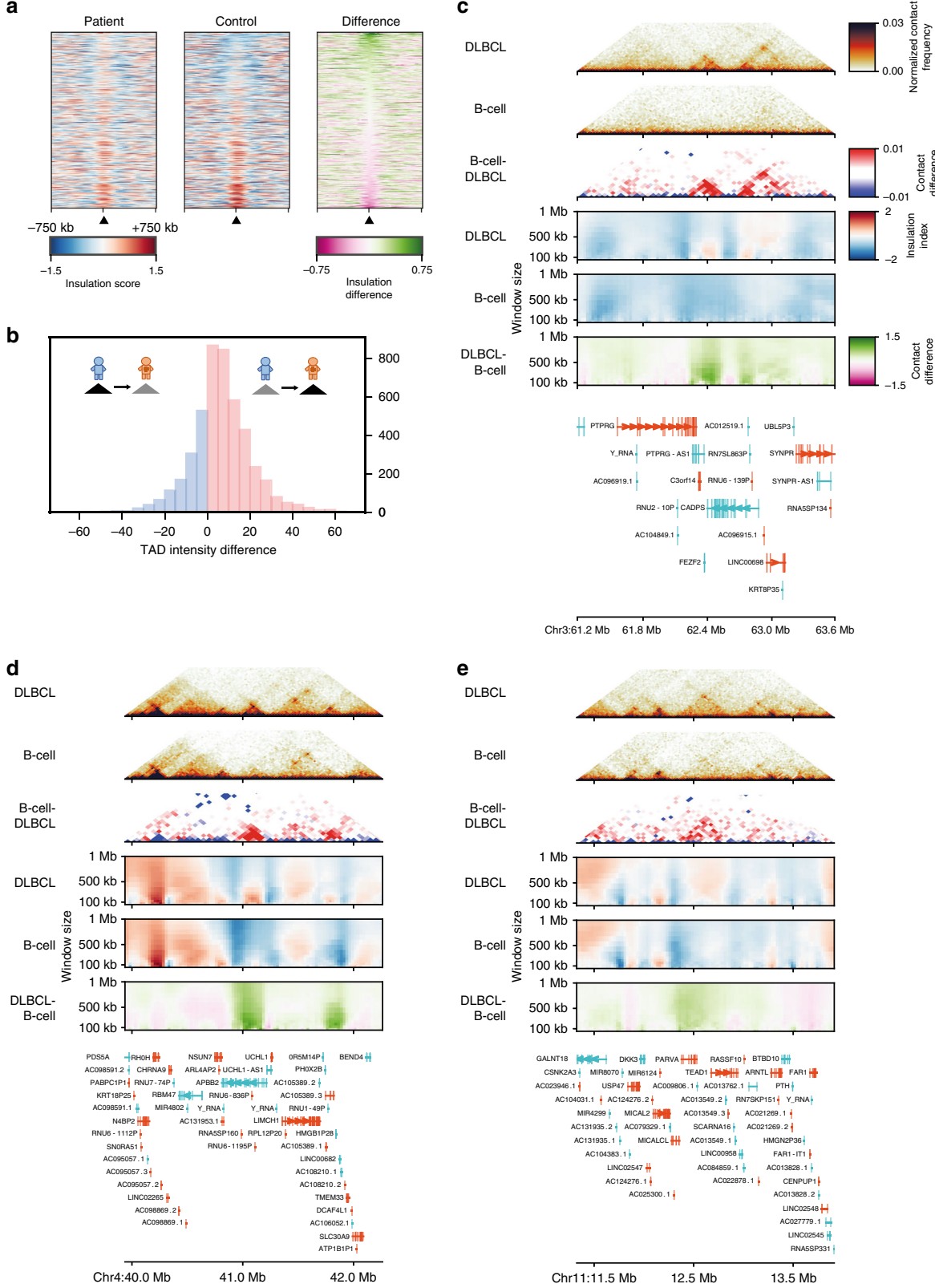

**Fig. 6** Extensive rewiring of chromatin organisation in DLBCL cells. **a** Insulation score changes between DLBCL (patient) and B-cell (control) for all regions in-between two consecutive TAD boundaries (centre marked by triangle). **b** Histogram of the differences in average Hi-C signal between two consecutive TAD boundaries from patient to control. **c–e** Examples of de novo TADs emerging in DLBCL showing (from top to bottom: Local DLBCL Hi-C, local B-cell Hi-C, difference in local Hi-C (DLBCL – B-cell), insulation scores for different window sizes in DLBCL, insulation score in B-cell, difference in insulation score (DLBCL – B-cell), genes in the region (GENCODE)

(5 M to 100 k samples) or 10 μL (50 k to 1 k samples) of 20 mg/mL proteinase K (Applichem) were added followed by 30 min incubation at 55 °C with shaking (1000 rpm). Subsequently, 130 μL (5 M to 100 k samples) or 65 μL (50 k to 1 k) of 5 M sodium chloride was added to the samples, which were then incubated overnight at 65°C with shaking (1000 rpm).

**DNA shearing**. DNA was precipitated by adding 630 μL (5 M to 100 k samples) or 315 μL (50 k to 1 k samples) of the Phenol-Chloroform-Isoamyl alcohol mix (25:24:1; Sigma-Aldrich). Samples were spun (5 min at 17,000 × $g$, 4 °C). The supernatant was then mixed to 63 μL (5 M to 100 k samples) or 31.5 μL (50 k to 1 k samples) of 3 M sodium acetate pH5.2, 2 μL GlycoBlue (Life Technologies) and 1008 μL (5 M to 100 k samples) or 504 μL (50 k to 1 k samples) pure ethanol, followed by 15 min incubation at −80 °C. Samples were then spun at 17,000 × $g$, 4 °C for at least 30 min. The supernatant was then discarded and the pellet was washed twice with ice-cold 70% ethanol (800 μL for 5 M to 100 k samples or 400 μL for 50 k to 1 k samples). All ethanol traces were removed, the pellet was then air-dried for no longer than 5 min and dissolved in 30 μL (5 M to 100 k samples) or 15 μL (50 k to 1 k samples) of Tris pH 8.0 (Applichem) and incubated for 5 min at 37 °C without rotation. RNA was digested by adding 1 μL RNAse (Applichem) to each sample and the mix was incubated for 15 min at 37 °C. Biotin from unligated fragments was removed by adding 10 μL (5 M to 100 k samples) or 5 μL (50 k to 1 k samples) of 10X NEB2 buffer (New England Biolabs), 1 mM of a dNTPs mix (2.5 μL for the 5 M to 100 k samples or 1.25 μL for the 50 k to 1 k samples), 20 mg/mL BSA (0.5 μL for 5 M to 100 k samples or 0.25 μL for 50 k to 1 k samples), nuclease-free water (up to 100 μL or 50 μL, respectively) and 3 U/μL T4 DNA polymerase (5 or 3 μL, respectively; New England Biolabs). Samples were mixed by gentle pipetting up and down and incubated at 20 °C for 4 h with no rotation. Samples were brought to a 120 μL volume with nuclease-free water and DNA was then sheared using a Covaris S220 instrument (2 cycles, each 50 s. long; 10% duty; 4 intensity; 200 cycles/burst).

**Biotin pull-down**. Dynabeads MyOne Streptavidin C1 beads (Thermo Fisher Scientific) were used to pull down biotinylated fragments. Briefly, 120 μL of the sheared Hi-C sample was mixed with an equal amount of the prepared magnetic beads and incubated for 15 min with rotation to allow DNA binding to the beads. The beads were washed twice with 1× B&W buffer + 0.1% Triton X-100 by incubating the samples at 55 °C with shaking (1000 rpm) for 2 min, followed by a wash with 10 mM Tris-Cl pH 8.0 and finally resuspended in 50 μL of the last wash solution.

**Preparation for illumina sequencing and final amplification**. The following steps were performed using reagents from the NEBNext Ultra DNA Library Prep Kit for Illumina (New England Biolabs). Briefly, libraries linked to the Dynabeads MyOne Streptavidin C1 were end repaired by adding 3 μL of the End Prep Enzyme Mix and 6.5 μL of the End Repair Reaction Buffer (10×), followed by an incubation in a thermocycler with the following programme: 20 °C for 30 min, 65 °C for 30 min, 4 °C hold. 15 μL of Blunt/TA Ligase Master Mix, 2.5 μL NEBNext Adaptor for Illumina and 1 μL of Ligation enhancer were added to the end-repaired sample and the mixture was incubated at 20 °C for 15 min in a thermocycler. Then, after the addition of 3 μL of USER enzyme the mixture was incubated at 37 °C for 15 min in a thermocycler. The beads were separated on a magnetic stand and washed twice using 1xB&W + 0.1% Triton X-100 and transferred to a 1.5 mL tube. A final wash with 10 mM Tris pH 8.0 was performed and beads were resuspended in 50 μL of the same solution.

A mock PCR was done to finely tune the number of cycles needed for library amplification. The number of cycles needed ranged from 12 to 20 PCR cycles (12 cycles: 5 M-HindIII, 100 k-HindIII, 1 M-MboI, 100 k-MboI, DLBCL, and B-cell control; 14 cycles: 50 k-MboI, 25 k-MboI and 10 k-MboI; 20 cycles for the 1 k-MboI sample). Each sample was individually barcoded (NEBNext Multiplex Oligos for Illumina, Index Primers Set 1) as follows: #4 (1 M-MboI), #6 (100 k-MboI), #1 (50 k-MboI), #3 (25 k-MboI), #5 (10 k-MboI), #7 (1 k-MboI), #4 (5 M- HindIII), #12 (100 k-HindIII), #10 (DLBCL), and #11 (B-cell control). Final amplification was done with four parallel reactions per sample as follows: 10 μL of the library bound to the beads, 25 μL of 2× NEBNext Ultra II Q5 Master Mix, 5 μL of 10 μM Universal PCR primer, 5 μL of 10 μM Indexed PCR primer and 10 μL of nuclease-free water. The PCR reactions were run using the following programme: 98 °C for 1 min, (98 °C for 10 s, 65 °C for 75 s, ramping 1.5 °C/s) repeated 12–20 times, 65 °C for 5 min, 4 °C hold.

After the amplification, the four reactions were combined into one tube, the volume was adjusted to 200 μL volume with nuclease free water, and then samples were size selected using Ampure XP beads (Beckman Coulter). Briefly, 0.55× volume of beads were mixed with the amplified sample and incubated at room temperature for 5 min. Samples were separated on a magnetic stand, supernatant was transferred to a new tube while beads (containing fragments too large to be sequenced) were discarded. 0.2× volumes of Ampure beads were then added to the sample and incubated at room temperature for 5 more min. After separating the beads on a magnetic stand, supernatant was discarded (containing fragments too short to be sequenced) and the beads were washed twice with 700 μL of 80% ethanol for 30 s each. Ethanol was completely removed and the beads were resuspended in 100 μL of 10 mM Tris-Cl pH 8.0 and incubated 1 min at room

temperature. To eliminate all traces of primers, 0.8× volumes of Ampure beads were added and the suspension was incubated for 5 min at room temperature. Beads were separated on a magnetic stand, the supernatant was discarded and the beads were washed twice with 700 μL of 80% ethanol, as before. All traces of ethanol were removed and the beads were resuspended in 25 (100 k to 1 k samples) or 50 μL (5 M and 1 M samples) of 10 mM Tris-Cl pH 8.0 and incubated at 37 °C for 5 min. Following separation, the supernatant contained the final in-situ Hi-C library. Libraries were quantified using Qubit dsDNA HS assay kit on a Qubit (Thermo Fisher Scientific), and using an Agilent DNA 1000 kit on a 2100 Bioanalyzer instrument (Agilent).

**Sequencing**. Samples were first pooled according to the restriction enzyme used and sequenced on an Illumina MiSeq (2 × 84 bp paired-end; MiSeq reagent kit v3-150 cycles) to assess library quality. Once the libraries were analysed and passed the quality criteria, namely, low number of inward/outward ligation errors, low PCR duplication levels, and low numbers of self-ligation errors (see below), they were sequenced on an Illumina NextSeq (2 × 80 bp paired-end; NextSeq 500/550 High Output kit v2-150 cycles).

**DLBCL and B-cell sample processing**. B-cells were obtained either from a blood extraction from a healthy donor or from a lymph node biopsy from a DLBCL (GC subtype) patient. The patient sample was obtained from the Department of Clinical Pathology at the Robert-Bosch-Hospital in Stuttgart (Germany) and its informed consent for retrospective analysis was approved by the ethics committee of the Medical Faculty, Eberhard-Karls-University and University Hospital Tübingen (reference no. 159/2011BO2). Control PBMCs were obtained from a healthy donor.

Control PBMCs were isolated from the in-between layer by density gradient centrifugation with Biocoll (Biochrom AG, Germany). PBMCs were resuspended in a mixture of 90% FBS (Gibco, Life Technologies, Germany) and 10% DMSO (Merck, Germany) and were snap-frozen at −80 °C for preservation. The patient sample came from a biopsy of a lymph node. Briefly, the biopsy was immediately cut into pieces, homogenised and resuspended generating a cell suspension. Cell suspensions were either directly used or frozen as unseparated cell (UC) populations in FBS and 10% DMSO. The snap-frozen samples were kept at -80°C, as previously described[52,53].

Once the samples were thawed, they were resuspended in RPMI media with 20% FBS, spun down to remove the remaining DMSO and then resuspended again in RPMI with 20% FBS. After this, cells were cross-linked in a 1% formaldehyde and quenched with 2.5 M Glycine solution, as previously described in this Methods section. A test to ensure that formaldehyde fixation will not affect the surface molecules was performed before and after fixation (Supplementary Figure 7). The viability of the surface molecules on a mixture of PBMCs from healthy donor and a mixed cell line population of HBL1 and Jurkat cells was assessed by staining with a CD19-PE (FL2; Miltenyi Biotech, 130-110-350, clone REA675, dilution 1:11) and CD20-FITC (FL1; Miltenyi Biotech, 130-091-108, clone LT20, dilution 1:11) antibodies (Supplementary Figure 8).

B-cells were then isolated by MACS-sorting[21] using a positive selection kit (Miltenyi Biotec, 130-091-104). Briefly, CD20$^+$ cells were labelled using magnetic coated CD20 MicroBeads, the cell suspension was loaded onto a MACS LS column (Miltenyi Biotec, 130-041-306) and placed on a magnetic field generated by a MACS Separator. The CD20$^+$ cells were retained into the column while the flow-through (unlabelled cells) was eliminated. Then the column was removed from the MACS Separator, the magnetically retained CD20$^+$ cells were then eluted and collected into a 15 mL Falcon tube. The performance of the MACS sorting was assessed by checking the B-cell presence and its proportions in the flow-through as well as in the eluted portion for the control PBMCs (Supplementary Figure 9a-b), for the mixed cell population sample (Supplementary Figure 9c-d) and for the patient sample (Supplementary Figure 10).

Once the eluted samples were recovered, we proceeded with the lysis and the rest of the Low-C library preparation as described above for the 50 k mESC sample.

**Bioinformatics processing of Low-C and Hi-C libraries**. Prior to mapping, the two mates of each paired-end reads sample were scanned for MboI ligation junctions, indicating sequencing through a Hi-C ligation product. If a junction was found, the read was split. Reads were then mapped independently to the *M. musculus* reference genome (mm10) using BWA-MEM (0.7.17), which may also result in split reads where the ends map to different locations in the genome. Those reads that did not align uniquely to the genome or that had a mapping quality lower than 3 were filtered out. Read pairs where one read was filtered out are discarded.

For the remaining read pairs, there are three possibilities: (i) none of the two reads in a mate pair was split in the pre-processing or mapping step (see above), (ii) one read in the pair was split, resulting in 3 mapped reads with the same ID, and (iii) one read in a pair was split multiple times or both reads were split at least once, resulting in more than 3 reads with the same ID. In case (iii) the mate pair is filtered out, as the exact interacting genomic location cannot be determined; in case (ii) the pair is considered valid if two reads map to the same genomic location (within 100 bp), otherwise it is discarded; case (i) is considered valid.

Restriction fragments in the genome were identified computationally using known restriction sequences of MboI and HindIII, and the remaining pairs of reads were assigned to the restriction fragments.

**Obtaining valid pairs of reads**. Pairs were filtered out if: (i) the mapped reads' distance to the nearest restriction site was larger than 5 kb, (ii) both reads mapped to the same fragment, or (iii) the orientation and distance of reads indicated a ligation or restriction bias[17,18]. Briefly, paired reads mapping in the same direction on the chromosome likely originate from a pair of fragments that had a cut restriction site between them and that had subsequently ligated—these were considered valid. Paired reads mapping in opposite directions may indicate that the reads map to a single large fragment with one or more uncut restriction sites. In this case, pairs facing inward would have originated from an unligated, pairs facing outward from a self-ligated fragment. At large genomic distances, there are approximately equal numbers of same and opposite orientation pairs. At shorter distances, there is an increased likelihood of uncut restriction sites between two reads, and pairs in opposite direction are filtered out. For every dataset, both the inward and outward ligation cut-offs have been fixed at 10 kb.

Finally, pairs were marked as PCR duplicates if another pair existed in the library that mapped to the same locations in the genome, with a tolerance of 2 bp. In those cases, only one pair from all duplicate ones for a given locus is retained for downstream processing. Finally, the genome was partitioned into equidistant bins and fragment pairs were assigned to bins using a previously described strategy[5]. The resulting contact matrix was filtered for low-coverage regions (with <10% of the median coverage of all regions) and corrected for coverage biases using Knight-Ruiz matrix balancing as described before[5,54]. Bins that had no contacts due to filtering were marked as 'unmappable'.

**Cis/trans ratio calculation**. The cis/trans ratio is calculated as the number of valid intra-chromosomal contacts (cis) to the valid inter-chromosomal contacts (trans). When comparing different species, this ratio will be affected by genome size and the number of chromosomes. We therefore also provide a 'species-normalised' cis/trans ratio by multiplying the trans value by the ratio of possible intra-chromosomal to inter-chromosomal contacts $f$ (the ratio of the number of intra-chromosomal pixels in the Hi-C map to the number of inter-chromosomal pixels).

**Observed/expected (OE) Hi-C matrix generation**. For each chromosome, we obtain the expected Hi-C contact values by calculating the average contact intensity for all loci at a certain distance. We then transform the normalized Hi-C matrix into an observed/expected (OE) matrix by dividing each normalized observed by its corresponding expected value.

**Aggregate TAD/loop analysis**. In general, average feature analysis is performed by extracting subsets of the OE matrix (can be single regions along the diagonal, or region pairs corresponding the matrix segments off the diagonal) and averaging all resulting sub-matrices. If the sub-matrices are of different size, they are interpolated to a fixed size using 'imresize' with the 'nearest' setting from the Scipy Python package.

TADs and loop anchors in Fig. 1 have been obtained from Rao et al.[5]. TADs and loop anchors in Fig. 4 have been called de novo from their respective datasets (see below). The region size for TADs has been chosen as 3× TAD size, centred on the TAD, and aggregate analyses have been performed in 25 kb matrices. The region size around loop anchors has been chosen as 400 kb in 25 kb matrices.

TAD strength is calculated as in Flyamer et al.[11]. Briefly, we calculate the sum of values in the OE matrix in the TAD-region and the sum of values for the two neighbouring regions of the same size divided by two. The TAD strength is then calculated as the ratio of both numbers.

Loop strength is calculated as in Flyamer et al.[11]. Briefly, we first calculate the sum of all values in the 300 kb region of the Hi-C matrix centred on the loop anchors. As a comparison, we calculate the same value for two control regions, substituting one of the loop anchors for an equidistant region in the opposite direction. The loop strength is then calculated as the original sum of values divided by the average sum of values in the two control regions.

**Expected values vs. distance**. Intra-chromosomal Hi-C matrix entries (50 kb resolution) were binned by distance to the diagonal and divided by the total number of possible contacts at each distance. The resulting average counts were plotted against distance in a log–log plot.

**AB compartments**. For each chromosome separately, the Hi-C matrix was converted to an OE matrix (see above). The OE matrix was then converted into a correlation matrix, where each entry $(i, j)$ represents the Pearson correlation between row $i$ and $j$ of the OE matrix. Finally, the signs of the first eigenvector entries were used to call compartments.

**Insulation score and TAD boundaries**. The insulation score was calculated as described before[19], by averaging contacts in a quadratic sliding window along the diagonal of the Hi-C matrix. Insulation scores were then divided by the chromosomal average and log2-transformed. Boundaries were calculated from the vector of insulation scores as previously described[19,42]. Aggregate TAD plots in Fig. 4, and the insulation and TAD intensity difference plots in Fig. 6 use the intervals between two consecutive boundaries as input.

**De novo loop calling**. Loops in the DLBCL and B-cell samples have been called using an in-house implementation of HICCUPS[5]. Briefly, for each entry in the Hi-C matrix, HICCUPS calculates several enrichment values over different local neighbourhoods (termed "donut", "lower-left", "horizontal" and "vertical" – for definition of the neighbourhoods see the original publication). Each enrichment value is associated with an FDR value for assessing statistical significance. We call loops at a matrix resolution of 25 kb and perform filtering exactly as described, only retaining loops that (i) are at least two-fold enriched over either the donut or lower-left neighbourhood, (ii) are at least 1.5-fold enriched over the horizontal and vertical neighbourhoods, (iii) are at least 1.75-fold enriched over both the donut and lower-left neighbourhood, and (iv) have an FDR ≤ 0.1 in every neighbourhood. We thus obtain 10,093 loops in the DLBCL and 13,213 loops in the B-cell samples —comparable to the number of loops identified originally in GM12878 cells[5].

**Identification of structural rearrangements in DLBCL**. To generate a list of candidate regions that may have undergone structural rearrangements in DLBCL, we performed Virtual 4C (V4C) for each Hi-C bin of the DLBCL matrix at 50 kb resolution (viewpoint), looking for peaks of signal away from the original viewpoint (target) that were not present in normal B-cells.

Specifically, in a Hi-C matrix $M$ of size $N \times N$, we examined each bin $i$, with $i \in [0, N]$. If any of the bins in the interval $[i - 7, i + 7]$ is unmappable (see above), it is not considered for further analysis, as we found that regions with mappability issues are typically false-positive rearrangements. We then obtained the vector $v$ of Hi-C signal as row i of M. The viewpoint peak height is then given by $v_i$. An entry $v_j$, with $j \neq i$, is considered a peak if it is larger than $0.15^*v_i$ and 99.5% of all other values in $v$ (the latter was introduced to filter out highly noisy V4C profiles). Peaks closer than 50 bins to $i$ are discarded as local enrichment of contacts.

V4C peaks are called as above for the DLBCL and the B-cell samples. We consider a peak as a putative rearrangement if it only occurs in the DLBCL, but not the B-cell sample. The final list of <100 putative rearrangements could then be inspected by eye in the local and inter-chromosomal Hi-C, eliminating highly noisy Hi-C regions and likely false-positives. Finally, this left just 14 peaks, of which 4 could be attributed to the ANXA3, and 10 to the t(3;14) rearrangements discussed in the manuscript.

**Hi-C difference matrices**. Plots highlighting differences between DLBCL and B-cell samples (Fig. 6) have been obtained by subtracting B-cell from DLBCL Hi-C matrices at 50 kb resolution. Pixels without signal in either datasets are removed for clarity.

**TAD intensity difference calculations**. To quantify the changes in TAD formation and intensity that occur from B-cell to DLBCL (Fig. 6a), we first merged boundaries in both samples (see above), and then calculated the average Hi-C signal between all possible pairs of contacts in-between two consecutive boundaries. This was done separately for the two datasets, and the TAD intensity difference for each region was calculated as the difference in average Hi-C signal of DLBCL and B-cell.

**Correlations**. All reported correlations are Pearson correlations. Corresponding plots were made using the "hexbin" plotting function on log-transformed counts from the matplotlib library version 2.0.0 in Python (matplolib.org).

The distance correlations in Fig. 2a have been obtained as follows: All intra-chromosomal contacts in a Hi-C map are first binned by distance. Bins are defined as [0–250 kb), [250 kb, 500 kb), [500 kb, 750 kb), … in the 50 kb resolution maps, [0–500 kb), [500 kb–1 Mb), [1.5–2 Mb), … in the 100 kb resolution maps, and [0–1 Mb), [1–2 Mb), [2–3 Mb), … in the 250 kb resolution maps. For each library (100 k, 10 k, 1 k, Dixon et al.[15], Du et al.[12]) correlations to the 1 M sample between all corresponding contact strengths in each bin are calculated. The x axis has been scaled to omit very large distances at which correlations become erratic due to the sparsity of the Hi-C matrix.

**Fluorescent in situ hybridisation analysis**. Interphase-FISH for *BCL6* (Vysis Break apart FISH probe kit, Abbot Molecular Diagnostics, Germany) was performed on 4 μm thick tissue sections cut from FFPE archival tissue blocks as previously described[55].

## Data availability

All relevant data supporting the key findings of this study are available within the article and its Supplementary Information files or from the corresponding author on reasonable request. The in situ Hi-C data generated in this study have been deposited in ArrayExpress under accession number E-MTAB-5875. Previously published Hi-C datasets used in this study are available in Gene Expression Omnibus (GEO; Rao et al.[5] GSE63525 ; Dixon et al.[15] GSE35156; Du et al.[12] GSE82185) and Genome Sequence Archive (GSA) (Ke et al.[13] PRJCA000241). Genome annotations have been downloaded from GENCODE, version 27.

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

## Acknowledgements

This research was funded by the Max Planck Society. We thank Caitlin MacCarthy and Hans Schöler for kindly providing the OG2 mouse stem cells (B6; CBA-Tg(Pou5f1-EGFP)2Mnn/J; stock number 004654) used in this experiment. We also thank Martin Stehling for assistance with fluorescence-activated cell sorting (FACS), and the Genomics core facility of the Medical Faculty Muenster for sequencing.

## Author contributions

J.M.V. conceived and supervised the study. N.D. performed in situ Hi-C experiments. K. K. performed computational analyses. T.E., G.O. and G.L provided clinical samples and assisted with B-cell isolation. A.S. performed FISH on clinical samples. N.D., K.K. and J. M.V. analysed and interpreted the data. N.D., K.K. and J.M.V. wrote the manuscript. All

authors participated in the discussion of the results, and commented on and approved the manuscript.

## Additional information

**Competing interests:** The authors declare no competing interests.

