## [Peer Review File · Nature Communications]

Reviewer #1 (Remarks to the Author):

In the manuscript (NCOMMS-17-18583b), the authors presented an improved Hi-C method for low input of cells which is highly reproducible and demonstrated that low-C is not affected by biases from the amount of cell number. However, two recently published papers had already generated very similar improved Hi-C methods using a small number of cells (Nature 544, 110–114 (2017) & Cell 170, 367–381.e20(2017)), making this manuscript with little novelty. Besides, there are only slight modifications in only a few steps in this protocol, without significant improvements compared with original in situ Hi-C methods. Decreasing library complexity has not been addressed in this research while this is the most difficult but important problem in the Hi-C methods using low amounts of input material. In addition, they didn't compare their results generated using this protocol with the results in two recently published papers either. Hence, this paper is not suitable for publication in Nature Communications.

Reviewer #2 (Remarks to the Author):

The authors describe Low-C, a method which is only subtly different from existing Hi-C methods and that is optimized for low cell numbers. The current state of the art is for Hi-C to be routinely used on populations of millions of cells, or for very specialized methods allowing single-cell Hi-C analysis, which carries its own problems in terms of what meaningful data can be obtained (many replicates are usually required; limited resolution, etc.). A "standard" Hi-C method which gives comparable performances in lower cell numbers is thus highly desirable when studying clinical samples or more complex biological processes. Low-C is the first described method which has a proper benchmarking compared to conventional methods, where it performs favorably. In its current state, I would support publication of the manuscript in a specialized or dedicated methods journal. However, I believe some extra information is required to give the method the true universality of use which would warrant publication in Nature Communications. These are feasible, so I request changes to the manuscript rather than suggest a rejection, and are highlighted below:

1. Low-C is essentially in-situ Hi-C with modifications optimized for lower cell numbers. To be a truly universal method, one needs to know to what extent these modifications are sufficient in more "challenging" cell/tissue types other than mESCs, or also need to be optimized. Although other cell/tissue types do not have published Hi-C data for benchmarking, assessment of PCR duplication events, ligation biases and ease of TAD identification should at least provide a proof of principle. For example, the authors could test Low-C on *Drosophila* imaginal disk and/or specific dissected mouse tissues (e.g. the developing limb buds used in papers cited within this manuscript).

2. Related to 1, it would be much more powerful to show a proof of principle of the utility of Low-C in answering a question usually not possible to address by conventional Hi-C. One example that springs to mind would be, if possible, to perform Low-C on a blood sample from a patient with Burkitt's lymphoma, and confirm the identification of the chromosome translocation. This is just a suggestion - other similar proofs of principle where translocations or some other chromosomal pathology can be identified by Low-C would work just as well.

Text edits:

3. 50 kb resolution is very nice for a Hi-C experiment, but of course is insufficient for identification of specific promoter-enhancer interactions. Unless the authors can demonstrate that Low-C can indeed detect specific chromatin loops, they should describe this limitation. Note that this limitation applies to all conventional Hi-C methods without very deep sequencing, so chromatin loops do not need to be identified by Low-C to warrant publication. It is just important that non-experts do not get unrealistic expectations of the method for their own studies.

4. A topic not covered in the manuscript is the issue of tissue heterogeneity, which may confound the use of Low-C on biopsy samples. This should be explicitly discussed.

5. In its current format, Table S1 is illegible.

6. On figure S1, what is the "400kb|2Mb|10Mb" text that appears below the contact heat maps?

7. When describing the PCA analysis, "strong clustering of Low-C" samples does indeed suggest reproducibility, but could also be interpreted as saying they are all systematically different to the conventional Hi-C samples, which did not cluster with them. This is not the case, and the figure is very convincing, but this text should be modified to avoid the confusion.

Reviewer #3 (Remarks to the Author):

The article "Low-C: An in situ Hi-C method for low numbers of cells" by Noelia Díaz et al. presents a derivative of the Hi-C technique to generate chromosome contact maps from small populations of cells.

The differences compared to the standard protocol consist essentially in changes in volume and timing, but also include an additional ~30 steps or so to the original protocol. These changes are nicely recapitulated in a table. The author compared contact maps generated from amount of cells as low as 1,000 cells with maps made with 1M cells, showing they can identify using low-C some of the 3D features of mouse chromosomes.

The work hold some potential, although I think the author could do a better job at analyzing the resolution limitation of the 1k cells assay. They should also discuss much more thoroughly what one could expect to recover regarding biologically relevant changes in the resulting low-C contact maps. Ideally, they should complete their proof of concept by performing low-C experiment on a truly limited sample with only a few thousand cells, and not only on a cell culture.

In Fig. 1b the authors show that the Pearson correlation is good between all datasets and control, but they do not mention the size of the bin they use. A look on Fig. 2a suggests it is actually a 250kb binning, which is quite high. It is unclear to me what the PC mean at this resolution. Even many failed Hi-C experiments eventually display high PC, because some of the signal close to the diagonal is so strong that it still emerge from the noise. Fig. 2a suggest PC at higher resolution is actually quite affected, with a PC ranging from 0.6 to 0.3 for one thousand cells. Therefore, the caveat mentioned in sentence 43 "Thus far, this restricts high-resolution analyses of population Hi-C to biological questions for which large numbers of cells are available [...]" remains in the low-C protocol, as the resolution remains, in my eyes, low (250kb). The authors do not really discuss the definition of resolution: that TADs and profile of contacts at increasing distances are conserved between experiments is not so surprising, as it is known that these parameters do not change much between cell types or experiments. Therefore, would a naive user find a great use in these results? This could be discussed and the authors should provide examples of significant changes in 50kb resolution (or 250kb) contact maps from the literature. The authors discuss contacts between promoters and enhancers, but as those are typically within the same TAD, changes at a low-C resolution are unlikely to be identified.

The main interest may be to detect chromosomal rearrangements (the citations could be improved to include recent work from the de Laat lab, among others) from limited tumor samples. However, the proof of concept experiment remains to be done, i.e. performing the low-C experiment on a real primary material.

Line 93: the author should cite Cournac et al. 2012 instead of or addition to Jin et al. 2013, as the filtering strategy described in the latter work was originally described in the former.

Point-by-point response for Diaz, Kruse et al., (NCOMMS-17-18583)

We would first like to thank the reviewers for showing interest in our work and for their constructive criticism and suggestions to improve the manuscript. In this revised version of the manuscript, we have addressed all of the reviewer comments, including the demonstration that Low-C can be directly applied in a clinical setup with primary tissue from patients. Given the updated message of the manuscript we have changed the title to: “Chromatin conformation analysis of primary patient tissue using a low input Hi-C method”.

In particular, upon revision, we have added the following datasets and analyses:

1. Low-C datasets at 25kb resolution for B-cells from a diffuse large B-cell lymphoma patient and normal B-cells from blood of a donor, as suggested by reviewers #2 and #3.
2. Evidence for the detection of chromosomal translocations and TAD rewiring in the patient’s sample.
3. *De novo* loop calls in the DLBCL and B-cell samples.
4. A more comprehensive comparison of Low-C to conventional Hi-C and other low input Hi-C methods.
5. A more thorough examination of Low-C library complexity.

Below, we provide a point-by-point response to the reviewers’ comments. To help with the structuring of this document, we have marked the reviewer’s comments with *blue italic typography*, while our replies are shown in normal black font. We have included in bold the main additions to the manuscript. We have numbered the reviewers’ comments when appropriate for extra clarity and to help cross-referencing similar comments.

Reviewer #1 (Remarks to the Author):

1.1. In the manuscript (NCOMMS-17-18583b), the authors presented an improved Hi-C method for low input of cells which is highly reproducible and demonstrated that low-C is not affected by biases from the amount of cell number. However, two recently published papers had already generated very similar improved Hi-C methods using a small number of cells (Nature 544, 110–114 (2017) & Cell 170, 367–381.e20(2017)), making this manuscript with little novelty. Besides, there are only slight modifications in only a few steps in this protocol, without significant improvements compared with original in situ Hi-C methods. Decreasing library complexity has not been addressed in this research while this is the most difficult but important problem in the Hi-C methods using low amounts of input material. In addition, they didn’t compare their results generated using this protocol with the results in two recently published papers either. Hence, this paper is not suitable for publication in Nature Communications.

We respectfully disagree with the reviewer’s evaluation of our work. In our original submission, we included a thorough evaluation of the performance of Low-C in a controlled dataset, which, as the rest of the reviewers agree, is necessary to be able to determine how comparable data generated by both methods are. This is of particular importance since some of the datasets in the Du et al. (2017)¹ and Ke et al. (2017)² studies display very little amounts of structure and, therefore, a thorough examination of the properties of these datasets is essential for the correct interpretation of these results.

Our original submission included analyses aimed at demonstrating that, despite the inherent decrease in library complexity due to PCR duplicates in the 1k sample, this still had enough complexity to generate comprehensive genome-wide chromatin maps at 50kb resolution. The final number of identified contacts in the 1k library (37.5M) is similar to those identified in the 1M library (45.0M). This is a significant improvement from the complexity generated in single-cell libraries that are usually very sparse and, at best, generate 0.48M contacts per genome³.

Upon revision, we have extended these analyses and now demonstrate that all Low-C libraries - down to 1k cells - have the potential to uncover the full complement of existing 3D contacts in the genome, and that the number of discovered unique pairwise contacts is a function of sequencing depth - not the input cell number (Supplementary Figure 2b-c). In addition, we now include two new Low-C datasets for healthy B-cells and diffuse large B-cell lymphoma (DLBCL) B-cells from human primary tissue sequenced at a high sequencing depth. These datasets demonstrate that the generated libraries have enough complexity to perform translocation detection analyses, compartment analysis, differential TAD analyses and loop detection analyses at 25kb resolution. In particular, our analyses uncover the presence of two chromosomal translocations and abundant changes in chromatin topology in the patient sample. To our knowledge, this is the first time that such analysis is performed at this level of resolution from primary patient material.

Finally, upon revision, and as suggested by the reviewer, we have also included the data from Du et al. (2017)¹ and Ke et al. (2017)² in our comparisons, showing a high degree of reproducibility between the methods.

Reviewer #2 (Remarks to the Author):

The authors describe Low-C, a method which is only subtly different from existing Hi-C methods and that is optimized for low cell numbers. The current state of the art is for Hi-C to be routinely used on populations of millions of cells, or for very specialized methods allowing single-cell Hi-C analysis, which carries its own problems in terms of what meaningful data can be obtained (many replicates are usually required; limited resolution, etc.). A "standard" Hi-C method which gives comparable performances in lower cell numbers is thus highly desirable when studying clinical samples or more complex biological processes. Low-C is the first described method which has a proper benchmarking compared to conventional methods, where it performs favorably. In its current state, I would support publication of the manuscript in a specialized or dedicated methods journal. However, I believe some extra information is required to give the method the true universality of use which would warrant publication in Nature Communications. These are feasible, so I request changes to the manuscript rather than suggest a rejection, and are highlighted below:

We thank the reviewer for the useful suggestions to improve the manuscript.

2.1. Low-C is essentially in-situ Hi-C with modifications optimized for lower cell numbers. To be a truly universal method, one needs to know to what extent these modifications are sufficient in more "challenging" cell/tissue types other than mESCs, or also need to be optimized. Although other cell/tissue types do not have published Hi-C data for benchmarking, assessment of PCR duplication events, ligation biases and ease of TAD identification should at least provide a proof of principle. For example, the authors could test

Low-C on Drosophila imaginal disk and/or specific dissected mouse tissues (e.g. the developing limb buds used in papers cited within this manuscript).

We agree with the reviewer's view. Rather than performing Low-C on developing tissue, we decided to take the "more powerful" proof of principle suggested by the reviewer in §2.2 and perform Low-C on primary material from clinically relevant samples (see below).

2.2. Related to 1, it would be much more powerful to show a proof of principle of the utility of Low-C in answering a question usually not possible to address by conventional Hi-C. One example that springs to mind would be, if possible, to perform Low-C on a blood sample from a patient with Burkitt's lymphoma, and confirm the identification of the chromosome translocation. This is just a suggestion - other similar proofs of principle where translocations or some other chromosomal pathology can be identified by Low-C would work just as well.

As suggested by the reviewer, we decided to use Low-C to generate high-quality, genome-wide libraries for B-cells for a DLBCL patient and control B-cells from blood from a donor. We decided to use these samples since: i) white blood cells from lymph node biopsies are used for a range of diagnostic procedures and hence it is desirable to use minimum amounts of cells; and ii) full ethical approval was available to use this material.

Following the reviewer's suggestions, we developed an analysis method for unbiased automatic detection of translocations in the patient data, revealing a t(3;14) translocation involving BCL6 and the IGH locus, which is a recurrent translocation in this type of disease. We validated the presence of the translocation in the patient sample using fluorescence in situ hybridisation (FISH; see new Figure 5i). In addition, we were able to detect a significant amount of variation at the TAD structure level, often overlapping the genomic location of previously identified lymphoma-related genes, suggesting that the regulation in these samples can be significantly different between disease and control.

Text edits:

2.3. 50 kb resolution is very nice for a Hi-C experiment, but of course is insufficient for identification of specific promoter-enhancer interactions. Unless the authors can demonstrate that Low-C can indeed detect specific chromatin loops, they should describe this limitation. Note that this limitation applies to all conventional Hi-C methods without very deep sequencing, so chromatin loops do not need to be identified by Low-C to warrant publication. It is just important that non-experts do not get unrealistic expectations of the method for their own studies.

We thank the reviewer for highlighting this important point, and we agree in full with the reviewer's view. We originally performed our analysis at 50kb resolution since this was a reasonable resolution given the limited amount of sequencing depth for our proof-of-principle mESC libraries. In this revised version of the manuscript, we include analyses at 25kb resolution for the newly generated datasets since these have been sequenced to a higher sequencing depth.

With respect to chromatin loops, we now show that loops can be visualised using aggregate loop analysis throughout the whole range of mESC libraries despite the limited sequencing depth. Note that this analysis is based on previously identified loops in more deeply sequenced datasets⁴ since the limited sequencing depth in these samples do not allow the

robust identification of *de novo* chromatin loops. To demonstrate that this is not a limitation of Low-C but rather a limitation of the amount of sequencing depth, we now show that chromatin loops can be detected *de novo* using 25kb resolution data in the DLBCL and normal B-cell samples that have been sequenced at a higher sequencing depth. These chromatin loops are clearly visible in our maps in Fig. 4b, 5d-e and Fig. 6c-e.

We agree with the reviewer that, in most cases, these loops will be mediated by architectural cohesin-CTCF binding events and only in some cases they might represent enhancer-promoter interactions. As the reviewer mentions, this is an intrinsic limitation of the Hi-C technology. We discuss this point in the revised discussion and we offer suggestions for potential alternative methods that will allow to detect these interactions using low amounts of input material.

2.4. A topic not covered in the manuscript is the issue of tissue heterogeneity, which may confound the use of Low-C on biopsy samples. This should be explicitly discussed.

We thank the reviewer for bringing up this point. Indeed, in order to minimise the tissue heterogeneity in our patient derived Low-C datasets, we performed magnetic cell sorting (MACS)⁵ of lymphocytes, to be able to specifically isolate the B-cell population (CD20+ cells). To do so, we first confirmed that: i) the fixation procedure did not affect the surface molecules needed for the CD20 magnetic beads to bind; ii) that the MACS sorting procedure efficiently isolates CD20+ cells; and iii) that this procedure has a similar efficiency in isolating cells from control and patient samples. We have included these results in Supp. Fig. 7-10, and in the corresponding sections of the results, discussion and methods sections.

Additionally, we are now discussing issues arising from tissue heterogeneity and sample composition specifically in the Discussion section.

2.5. In its current format, Table S1 is illegible.

We apologised for the inconvenience in accessing the information in Table S1. Table S1 is meant to be provided as a spreadsheet and the formatting seems adequate when we access it through appropriate software. Perhaps the reviewer only has access to this table through the PDF conversion, with the corresponding potential formatting issues. Otherwise, we would be happy to implement any specific suggestions from the reviewer regarding the format of the table.

2.6. On figure S1, what is the "400kb|2Mb|10Mb" text that appears below the contact heat maps?

We apologise for the lack of explanation in the figure legend. "400kb|2Mb|10Mb" correspond to the genomic distance between tick marks in the Hi-C maps, being 400kb the separation between minor ticks, 2MB the separation between major ticks, and 10Mb the size of the whole region represented in the plot. Originally, we added this labelling to the plots to give the reader a quick estimate of the size of each region without having to subtract the genomic coordinates. In this revised version, we have simplified the representation by only showing and labelling the major ticks.

2.7. When describing the PCA analysis, "strong clustering of Low-C" samples does indeed suggest reproducibility, but could also be interpreted as saying they are all systematically

different to the conventional Hi-C samples, which did not cluster with them. This is not the case, and the figure is very convincing, but this text should be modified to avoid the confusion.

We thank the reviewer for this suggestion. We have modified the text accordingly.

Reviewer #3 (Remarks to the Author):

The article “Low-C: An in situ Hi-C method for low numbers of cells” by Noelia Díaz et al. presents a derivative of the Hi-C technique to generate chromosome contact maps from small populations of cells. The differences compared to the standard protocol consist essentially in changes in volume and timing, but also include an additional ~30 steps or so to the original protocol. These changes are nicely recapitulated in a table. The author compared contact maps generated from amount of cells as low as 1,000 cells with maps made with 1M cells, showing they can identify using low-C some of the 3D features of mouse chromosomes.

3.1. The work hold some potential, although I think the author could do a better job at analyzing the resolution limitation of the 1k cells assay.

We thank the reviewer for the interest in our work. Upon revision we have expanded the analyses that we perform in the proof-of-principle comparison presented in the manuscript. In particular, we now include a systematic analysis of compartments, distance decay, insulation score analysis, aggregate TAD analysis and aggregate loop analysis. The limited amount of sequencing depth for the mESC libraries do not allow us to perform analyses at more than 50kb resolution for these samples. However, we show that the limitation is similar for all samples and we now provide formal evidence that this limitation is due to sequencing depth rather than to library complexity in the 1k samples.

3.2. They should also discuss much more thoroughly what one could expect to recover regarding biologically relevant changes in the resulting low-C contact maps.

We thank the reviewer for this comment. In the revised version of the work we now include Low-C chromatin contact maps for a limited cell sample of DLBCL and control B-cells at 25kb resolution. In addition to being able to detect genome rearrangements in these samples, we are now able to perform a systematic comparison between healthy and disease cells, which revealed a significant amount of chromatin contact changes at the level of TADs. We have included and discussed these results in the revised version of the manuscript.

3.3. Ideally, they should complete their proof of concept by performing low-C experiment on a truly limited sample with only a few thousand cells, and not only on a cell culture.

As mentioned above in response to comment §2.2 from reviewer #2, and as requested by reviewer #3, we have now generated Low-C maps for CD20+ cells (B-cells) from primary tissue from a DLBCL patient and blood from a healthy donor as a control, demonstrating the applicability of the method to clinical cases with truly limited availability of input material.

3.4. In Fig. 1b the authors show that the Pearson correlation is good between all datasets and control, but they do not mention the size of the bin they use. A look on Fig. 2a suggests it is actually a 250kb binning, which is quite high. It is unclear to me what the PC mean at this

resolution. Even many failed Hi-C experiments eventually display high PC, because some of the signal close to the diagonal is so strong that it still emerge from the noise.

We apologise for the lack of clarity and the omission of information in the previous submission. The Pearson correlation (PC) that we calculated in Fig.1 was obtained using the maps displayed in the same figure, which were produced at 50kb resolution. As mentioned above, please note that these plots are produced from libraries with limited sequencing depth (24-43*10⁶ reads), which do not allow us to perform robust analyses at higher resolution. Also, please note that this is not a limitation of the method *per se*, but rather an experimental design decision. We decided not to sequence these samples at higher resolution given the availability of other publicly available mESC datasets sequenced at a higher sequencing depth.

We agree with the reviewer that “whole map” correlation coefficients can be strongly affected by the signal along the diagonal. To minimise this confounding factor, we have calculated the correlation coefficients of contacts at increasing ranges of distances away from the diagonal (Fig. 2a). Please see the response below (§3.6) where we specifically address this point.

3.6. Fig. 2a suggest PC at higher resolution is actually quite affected, with a PC ranging from 0.6 to 0.3 for one thousand cells. Therefore, the caveat mentioned in sentence 43 “Thus far, this restricts high-resolution analyses of population Hi-C to biological questions for which large numbers of cells are available [...]” remains in the low-C protocol, as the resolution remains, in my eyes, low (250kb).

We thank the reviewer for this comment, that brought to our attention the need to clarify the message for Fig. 2a. The aim for this plot (as well as for Figs. 1-3) is to demonstrate the robustness of the obtained libraries upon titration of the starting material, as well as to show that the resulting libraries are consistent with other published Hi-C datasets for the same cell type.

The plot in Fig. 2a, shows the correlation levels with the 1M cells sample at increasing genomic distances at three different levels of resolution (50kb binning, 100kb and 250kb). The decrease in correlation observed at increasing linear distances is expected for any Hi-C dataset given the reduction in measured contacts between more distant loci that is characteristic of these datasets, including the Low-C datasets presented here (Supp. Fig. 5). To demonstrate that Low-C datasets do not display an associated bias due to the reduced amount of input material, we now include in the plot a calculation of the correlation with two previously published mESC Hi-C datasets (from Dixon et al. (2012)⁶ and Du et al. (2017)¹), which show no differences to the Low-C datasets. In addition, we repeated this analysis on subsampled versions of these datasets so they all have equal number of valid read pairs (Supp. Fig 2a), which resulted in similar or better correlation values for Low-C datasets. These results demonstrate that, as with conventional Hi-C, the resolution of the Hi-C map is primarily dependent on the sequencing depth.

We hope that this clarifies the reviewer concerns regarding the correlation analyses presented in the manuscript.

3.7. The authors do not really discuss the definition of resolution: that TADs and profile of contacts at increasing distances are conserved between experiments is not so surprising, as it is known that these parameters do not change much between cell types or experiments.

Therefore, would a naive user find a great use in these results? This could be discussed and the authors should provide examples of significant changes in 50kb resolution (or 250kb) contact maps from the literature.

We are slightly unsure about this comment by the reviewer. Our proof-of-principle experiment was designed to test the robustness and reproducibility of the method when scaling it down to low amounts of input. Since the input material (mESC) was the same cell type for all conditions, a successful experiment when titrating the amount of input material would show no change in TAD and contact profiles until the amount of technical variability (e.g., PCR amplification biases, inefficient chromatin digestion or ligation, etc). Despite the moderate amount of sequencing depth, we do not see such differences, either in the maps or in the measurements that are usually derived from these (components, insulation score, loops, etc), which demonstrates that the method is robust and reproducible. As such, our results are important since they offer proof-of-concept regarding the quality of the data generated using low amounts of input material.

It might be that the reviewer's concern was whether Low-C datasets had enough resolution to detect changes at the TAD organisation level. In order to address this point, rather than performing a comparison between publicly available datasets and our mESC Low-C datasets (since that analysis would mix confounding technical differences, and it is unclear which dataset should be used to make such comparison), we used our newly generated Low-C datasets for DLBCL and healthy B-cells (see point §2.2 above) to determine whether we could detect changes in TAD structure between samples. Our new results included in Fig. 6 and Supp. Table 4 show that there is a significant amount of variation at TAD level between the samples, demonstrating that Low-C datasets do have the resolution to detect these changes.

3.9. The authors discuss contacts between promoters and enhancers, but as those are typically within the same TAD, changes at a low-C resolution are unlikely to be identified.

We agree with the reviewer and have updated the text accordingly. As we mention in §2.3 above, it should be noted though, that we are able to call chromatin loops *de novo* from Low-C data at 25kb resolution (see Fig. 4 in the revised manuscript), some of which overlap with promoters. Nevertheless, we agree with the reviewer and we have revised our discussion to offer suggestions for potential alternative methods that will allow to detect these interactions using low amounts of input material.

3.10. The main interest may be to detect chromosomal rearrangements (the citations could be improved to include recent work from the de Laat lab, among others) from limited tumor samples. However, the proof of concept experiment remains to be done, i.e. performing the low-C experiment on a real primary material.

We thank the reviewer for this comment, which is echoed by a similar suggestion from reviewer #2. As explained above (§2.2), in the revised version of the manuscript we have included Low-C datasets generated from primary tissue from a DLBCL patient and B-cells from a healthy donor as control. We then use these new data to perform an unbiased detection of potential translocations in the patient's sample, revealing a reciprocal t(3;14) translocation involving BCL6 and the IGH locus, a hallmark of this type of tumours. Furthermore, we perform a characterisation of the level of TAD structure variation, revealing a significant

amount of TAD gains overlapping with genes previously identified as relevant in these pathologies.

We have also updated the citations as suggested by the reviewer. Specifically, we now include:

Simonis, M. *et al.* High-resolution identification of balanced and complex chromosomal rearrangements by 4C technology. *Nat. Methods* **6**, 837–842 (2009).

Krijger, P. H. L. & de Laat, W. Regulation of disease-associated gene expression in the 3D genome. *Nat. Rev. Mol. Cell Biol.* **17**, 771–782 (2016).

Harewood, L. *et al.* Hi-C as a tool for precise detection and characterisation of chromosomal rearrangements and copy number variation in human tumours. *Genome Biol.* **18**, 125 (2017).

Lin, D. *et al.* Digestion-ligation-only Hi-C is an efficient and cost-effective method for chromosome conformation capture. *Nat. Genet.* **50**, 754–763 (2018).

van de Werken, H. J. G. *et al.* Robust 4C-seq data analysis to screen for regulatory DNA interactions. *Nat. Methods* **9**, 969–972 (2012).

Zepeda-Mendoza, C. J. *et al.* Quantitative analysis of chromatin interaction changes upon a 4.3 Mb deletion at mouse 4E2. *BMC Genomics* **16**, 982 (2015).

3.12. Line 93: the author should cite Cournac et al. 2012 instead of or addition to Jin et al. 2013, as the filtering strategy described in the latter work was originally described in the former.

We thank the reviewer for this remark. We have updated the citation accordingly and we now cite both works, as we are also using the specific representation used in Jin et. al. (2013)⁷.

References

1. Du, Z. *et al.* Allelic reprogramming of 3D chromatin architecture during early mammalian development. *Nature* **547**, 232–235 (2017).
2. Ke, Y. *et al.* 3D Chromatin Structures of Mature Gametes and Structural Reprogramming during Mammalian Embryogenesis. *Cell* **170**, 367–381.e20 (2017).
3. Lando, D., Stevens, T. J., Basu, S. & Laue, E. D. Calculation of 3D genome structures for comparison of chromosome conformation capture experiments with microscopy: An evaluation of single-cell Hi-C protocols. *Nucleus* **9**, 190–201 (2018).
4. Rao, S. S. P. P. *et al.* A 3D map of the human genome at kilobase resolution reveals principles of chromatin looping. *Cell* **159**, 1665–1680 (2014).
5. Yan, H. *et al.* Magnetic cell sorting and flow cytometry sorting methods for the isolation and function analysis of mouse CD4⁺ CD25⁺ Treg cells. *J. Zhejiang Univ. Sci. B* **10**, 928–32 (2009).
6. Dixon, J. R. *et al.* Topological domains in mammalian genomes identified by analysis of chromatin interactions. *Nature* **485**, 376–380 (2012).
7. Jin, F. *et al.* A high-resolution map of the three-dimensional chromatin interactome in human cells. *Nature* **503**, 290–4 (2013).

Reviewer #2 (Remarks to the Author):

The authors have satisfactorily replied to my suggestions, and I fully support publication in Nature Communications.

Reviewer #3 (Remarks to the Author):

First, I would like to apologize to the authors for the delay in reviewing this revised version of their manuscript.

I have now had the time to read their detailed response as well as the revised manuscript.

I appreciated the inclusion of the data of Du et al and Ke et al for comparison with the present samples, which as pointed out by the referee 1 were much needed. I recommend the author to avoid overstating the novelty of this work with respect to these former publications (including on social media), but to insist more on the benchmarking they did with respect to low C results. The clarifications regarding the quality control of the results are now satisfying. The discussion about the limitations of the LowC approach is also much more clear. The inclusion of LowC data on B cells from a DLBCL patient also make the article much more relevant to a broad audience.

I do not have major concern at this stage regarding the publication of this work.

Response to Reviewers

We would first like to thank the reviewers for their feedback. Here, we provide a point-by-point response to the reviewers' comments. To help with the structuring of this document, we have marked the reviewer comments with *blue italic typography*, while our replies are shown in normal black font.

Reviewer #2 (Remarks to the Author):

The authors have satisfactorily replied to my suggestions, and I fully support publication in Nature Communications.

We thank the reviewer for the former revision and suggestions to improve the manuscript.

Reviewer #3 (Remarks to the Author):

First, I would like to apologize to the authors for the delay in reviewing this revised version of their manuscript. I have now had the time to read their detailed response as well as the revised manuscript.

I appreciated the inclusion of the data of Du et al and Ke et al for comparison with the present samples, which as pointed out by the referee 1 were much needed. I recommend the author to avoid overstating the novelty of this work with respect to these former publications (including on social media), but to insist more on the benchmarking they did with respect to low C results. The clarifications regarding the quality control of the results are now satisfying. The discussion about the limitations of the LowC approach is also much more clear. The inclusion of LowC data on B cells from a DLBCL patient also make the article much more relevant to a broad audience. I do not have major concern at this stage regarding the publication of this work.

We thank the reviewer for the comments and suggestions. We thoroughly re-read the manuscript to ensure that we had not overstated the novelty of Low-C with respect to other low input Hi-C methods, and we acknowledge the advice to do the same on social media.